

# Organization of Dust Storms and Synoptic Scale Transport of Dust by Kelvin Waves

**Ashok Kumar Pokharel[1, 2] and Michael L. Kaplan[2, 3]**

5    [1]Department of Atmospheric Sciences /University of Nevada, Reno, USA.

[2]Division of Atmospheric Sciences/Desert Research Institute, Reno, NV, USA.

[3]Applied Meteorology Program/ Embry- Riddle Aeronautical University, Prescott, AZ, USA

Corresponding author: Ashok Kumar Pokharel (ashokpokharel@hotmail.com)



**Abstract**

Based on the large scale transport of dust driven by the winds parallel to the mountains in the Harmattan, Saudi Arabian, and Bodélé Depression dust storms cases, a detailed study of the generation of Kelvin Waves and its possible role in organizing these dust storms and large scale dust transport was accomplished. For this study, observational and numerical model analyses were done in an in depth manner. For this, MERRA reanalysis datasets, WRF simulated high resolution variables, MODIS/Aqua and Terra images, EUMETSAT images, NAAPS aerosol modelling plots, and MERRA-2 dust scattering AOD modelling plots, surface observations, and rawinsonde soundings were analyzed for each of these three case studies. We found there were meso-β scale (horizontal length scale of 20-200 km) adjustment processes resulting in Kelvin waves only in the Harmattan and the Bodélé Depression cases. The Kelvin wave preceded a cold pool accompanying the air behind the large scale cold front instrumental in the major dust storm. We find that this Kelvin wave organized the major dust storm in a narrow zone parallel to the mountains before it expanded upscale (meso-α to synoptic).

## 1. Introduction

The Sahara Desert is dominated most of the year by the northeasterly wind-driven dry and hot air originating from the anticyclonic system centered over the North Atlantic Ocean (Shao, 2000). It has been shown that the transport of dust from the Sahara takes place in discrete outbreaks of several days and outbreak location and direction changes with the fluctuations of the Intertropical Convergence Zone (ITCZ) seasonally (Swap et al., 1996; Moulin et al., 1998). The annual maximum dust storm frequency and intensity in the Sahara usually takes place in late winter and spring (Swap et al., 1996).

In Africa there are two preferred dust source regions – the first comprises Algeria, Mauritania, and Morocco from which dust is transported over the Atlantic and as far west as the Barbados Islands and the second is the Chad Basin which exports dust to countries around the Gulf of Guinea (Balogun, 1974). The Bodélé Depression which lies in the Chad Basin is the most intense and perpetual dust source in the world followed by the Western Sahara and has become the biggest and most persistent dust emission source on the global scale because of the availability of large amount of deflatable material and strong wind systems to facilitate the long range transport of the material (Engelstaedter et al., 2006). The emission of dust from Bodélé takes place throughout the year though the peak emission occurs in winter/spring (Washington and Todd, 2005; Washington et al., 2006). These facts indicate the importance of the study of the Harmattan and Bodélé dust storms. Studies of these phenomena have been increasing since these two systems deliver a significant amount of dust equatorward, causing severe negative impacts on visibility, agriculture, and human health (e.g., causes meningitis) (Burton et al., 2013; Kalu, 1979; Pérez García-Pando et al., 2014).

Alharbi et al. (2013) state that dust storms in Saudi Arabia are very frequent. Washington et al. (2003) indicate that eastern and central Saudi Arabia are also areas of dust storm activity. The primary local source of dust storm activity in Saudi Arabia is the Rub Al Khali. Additional more remote sources are the Saharan Desert for Western Saudi Arabia and Iraqi deserts for the northern and eastern part of Saudi Arabia (Notaro et al., 2013). These frequent dust storms in Saudi Arabia and the arid and semiarid areas around the Arabian Sea are some of the most important global dust sources (Kutiel and Furman, 2003). Hamidi et al. (2013) mentioned that the Tigris-Euphrates alluvial plain was the major dust source in the Middle East. Alharbi et al.,





(2013) mentioned that the severe dust storm of March 10, 2009 was one of the most intense dust storms experienced in Saudi Arabia over the last two decades period.

All three severe dust storms: 1) Harmattan dust storm of March 2, 2004 in north West Africa, 2) Saudi dust storm of March 9, 2009, and 3) Bodélé Depression dust storm of December 8, 2011, that occurred in the lee of the mountains were caused by downslope wind effects and jet adjustment processes (Pokharel et al. 2017a and 2017b). During these events the winds were also parallel to the lee of the respective mountains (Atlas Mountains in the Harmattan case, Sarawat Mountains in the Saudi case, and Tibesti Mountains in the Bodélé case). The location of strong dynamics for these case studies is shown by the Modern Era Retrospective-Analysis for Research and Applications (MERRA) analysis data sets. Over time the wide distribution of dust ablated and transported away from the mountains in each of these cases indicated that a possible role of terrain-induced waves such as Kelvin waves may be important in transporting dust from these three severe dust storms events. For this study, it was hypothesized that first organization of dust storms and spread of dust into the larger scale circulations was facilitated by Kelvin waves (the generation of Kelvin wave is illustrated by a schematic diagram in Figure 1). In this study, we have pursued an evaluation of the possible role of Kelvin waves in the organization of strong dust transport events. The next section includes a literature review of Kelvin waves followed by research methodology, results and discussion, and conclusion in our analyses of dust storm genesis and dust transport.

## 1.1  Relevant theories of Kelvin waves

Wang (2002) stated that a Kelvin wave is a relatively long wavelength gravity wave affected by the earth's rotation that is trapped along a lateral boundary, e.g., mountain ranges or coastlines. The generation and type of the Kelvin wave relies on the relative importance of the restoring force of gravity accompanying stable stratification, the significance of Coriolis acceleration, and the nature of the physical boundary including the possible role of the proximity of the equator. To create a Kelvin wave in this study's region of interest, e.g., the Atlas Mountains in northwest Africa, the following processes (schematic Figure 1)  must occur 1) early on the shallow cold pool of air is blocked by the Atlas Mountains due to lack of sufficient kinetic energy to force air parcels over the mountain, causing a buildup of mass as a height perturbation in the area immediately adjacent to it, 2) after a certain period of time < than an inertial period ($<2\pi/f$ where $f$ is the Coriolis parameter) this build-up of excess mass was released as a gravity wave or buoyancy wave signal or undulation in a free surface near the terrain (Thomson, 1879), 3) as time progresses towards $2\pi/f$ the Coriolis force acts on and turns the flow towards the right in a very slow effort to achieve balance, 4) as this process continued wind flow became parallel to the mountains as a distinct wind perturbation in conjunction with the aforementioned free surface undulation, and 5) in time the wind accompanying this Kelvin wave accelerates parallel to the mountain (Tilley, 1990).

## 2.  Materials and Methods

To study these three dust events in detail, we have used Meteosat-8 dust image captured from the European Organization for the Exploitation of Meteorological Satellites (EUMETSAT) (http://www.eumetsat.int/website/home/Images/ImageLibrary/DAT_IL_04_03_06_F.html),a composite of the  Moderate Resolution Imaging Spectroradiometer (MODIS) /Aqua and Terra (level 1b, collection 51, 1 km horizontal resolution, and RGB composite)



**Figure 1.** Schematic diagram shows the blocking of the flow, height perturbation, excess mass release, acting of Coriolis force and turning of the flow towards the right to achieve a balance, wind flow accompanying Kelvin waves propagating along the mountains, interaction of this wind accompanying Kelvin waves with the warm air column resulting in significant turbulence kinetic energy (TKE) leading to dust storm.

(a) D (Fluid depth) (Fluid blocked by the mountain) — Mountains — X

(b) D (Fluid depth), height perturbation — Mountains — X

(c) Release of excess mass after certain period of time as a gravity wave — Mountains — X, Y

(d) Coriolis force (fv) acts on and turns the flow towards the right — Mountains — X, Y

(e) Wind flow accompanying Kelvin waves parallel to the mountains — Mountains — X, Y

(f) Dust emission — Warm air in the leeside — Interaction of wind flow accompanying Kelvin waves with the highly warm air column created significant magnitudes of TKE resulting dust from the surface — Mountains — X, Y

$$\frac{\partial TKE}{\partial t} = -V.\nabla TKE + u*^2 \frac{\partial u}{\partial z} + \frac{g}{Tv}Qs - \varepsilon$$



(https://ladsweb.nascom.nasa.gov) (Figure 2), and aerosol optical depth (AOD) derived from the MODIS/Aqua instrument (level 3 daily (D3)). Rawinsonde soundings obtained from the University of Wyoming and observational surface data from Weather Underground were also analyzed. The low resolution evolution of the dust storms are studied with an aerosol-weather model simulation of 1º X 1º horizontal resolution from the Navy Aerosol Analysis and Prediction System (NAAPS) (https://www.nrlmry.navy.mil/aerosol_web/). Besides this, the Modern Era Retrospective-Analysis for Research and Applications-2 (MERRA- 2) model (spatial resolution 0.5 X 0.625 º) hourly data sets were used to analyze the dust scattering (AOD) 550 nm of 1 μm particulate matter (PM) over this region of interest (https://giovanni.gsfc.nasa.gov/giovanni/).

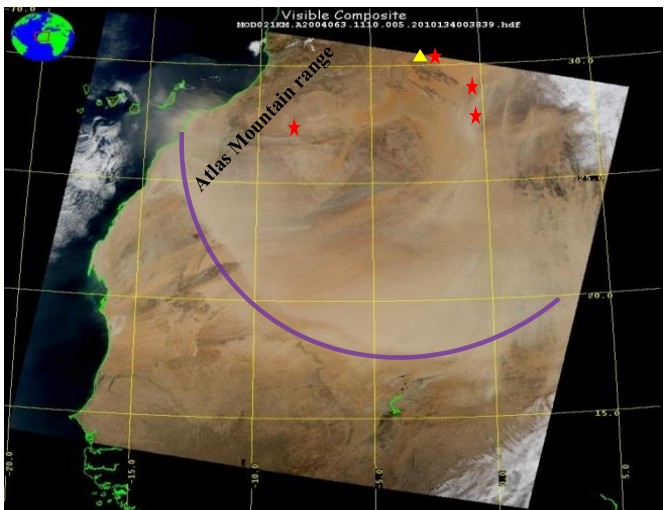

**Figure 2:** Dust storm image captured by MODIS/AQUA at 1110 UTC on March 3, 2004 (source: https://ladsweb.nascom.nasa.gov).The purple colored arc depicts the outline of the arc of dust plume. The red stars indicate surface stations (Bechar, Tindouf, Timimoun, and Adrar ) in Algeria of which x and y coordinates are mentioned in sub-section of observational and model subsection 3.1.1 of section 3.1 Harmattan dust storm case study) which captured the dust storms on 2 March 2004 (source: wunderground.com). The yellow triangle indicates soundings station at Bechar in Algeria.

For the study of synoptic to meso-scale observational atmospheric processes that organize conditions favorable for a Kelvin wave, surface pressure, geopotential height, air temperature, wind speed and direction, vertical motion, vertically integrated atmospheric mass tendency, and kinetic energy due to the pressure gradient force obtained from the Modern Era Retrospective-Analysis for Research and Applications (MERRA) model (https://disc.sci.gsfc.nasa.gov/mdisc/data-holdings/merra/merra_products_nonjs.shtml; Rinecker et al. 2011) were analyzed. These data were used to create horizontal cross sections at different pressure levels as well as vertical cross sections of u and v wind speed, and isentropic surfaces at a resolution of 0.50º X 0.67º.

In order to get finer temporal and spatial resolution of the atmospheric processes, which were involved in the dust storms, non-hydrostatic Weather Research and Forecasting (WRF) model (Skamarock et al., 2008) were run for each dust storm, given in Table 1. Four nested domains with horizontal resolutions of 54, 18, 6, and 2 km were generated, of which resolutions higher than 18 km were run with moist convection turned off as a parameterization because convection was virtually nonexistent in these cases. The lateral boundary condition data of the parent domain with the lowest resolution were from the National Center for Environmental Prediction (NCEP)/Global Forecasting System (GFS) (1° X 1°) products. Three domains were then nested into the parent domain having 18, 6, and 2 km. The WRF simulated domains for the three cases are presented in Figures 3, 10, and 13. The following six physical parameterizations, such as (i) momentum and heat fluxes at the surface (Janjić, 1996, 2001) following Monin-



Obukhov similarity theory, (ii) turbulent mixing following the Mellor-Yamada-Janjić 1.5 order
(level 2.5) turbulence closure model (Mellor and

| Dust storm cases | List of observational data sets | Horizontal grid dimensions (west-east and north-south directions) | Integrated time | WRF model physics |
|---|---|---|---|---|
| Harmattan dust storm | MODIS and EUMETSAT imageries, radiosonde soundings, aerosol optical depth by MODIS, surface data from Weather Underground, and MERRA data sets | 82 X 118 grid points (54 km grid spacing) | 1200 UTC March 1 to 1800 UTC March 3, 2004 | 1. Momentum and heat fluxes at the surface using an Eta surface layer scheme [*Janjić*, 1996, 2001] that follows Monin-Obukhov similarity theory, |
| | | 208 X 274 grid points (18 km grid spacing) | 1800 UTC March 1 to 1800 UTC March 3, 2004 | |
| | | 502 X 613 grid points (6 km grid spacing) | 0000 UTC March 2 to 1800 UTC March 3, 2004 | 2. Turbulence processes following the Mellor-Yamada- Janjić 1.5 order (level 2.5) turbulence closure model [*Mellor and Yamada,* 1974; *Janjić,* 2001], |
| | | 802 X 802 grid points (2 km grid spacing) | 1200 UTC March 2 to 1800 UTC March 3, 2004 | |
| Saudi Dust storm | MODIS satellite image, radiosonde soundings, surface data from Weather Underground, and MERRA data sets | 82 X 97 grid points (54 km grid spacing) | 0000 UTC March 9 to 1200 UTC March 10, 2009 | 3. Convective processes following the Betts-Miller-Janjić cumulus scheme [*Betts,* 1986; *Betts and Miller,* 1986; *Janjic,* 1994]- applied only on the 54 and 18 km grid, |
| | | 208 X 232 grid points (18 km grid spacing) | 0600 UTC March 9 to 1200 UTC March 10, 2009 | |
| | | 502 X 553 grid points (6 km grid spacing) | 1200 UTC March 9 to 1200 UTC March 10, 2009 | 4. Cloud microphysical processes following the Thompson double-moment scheme [*Thompson et al.,* 2004, 2006], |
| | | 802 X 760 grid points (2 km grid spacing) | 1200 UTC March 9 to 1200 UTC March 10, 2009 | |
| Bodélé dust storm | MODIS image, surface data from Weather Underground, and MERRA data sets | 82 X 97 grid points (54 km grid spacing) | 0000 UTC December 8 to 1200 UTC December 9, 2011 | 5. Radiative processes following the Rapid Radiative Transfer Model for long wave radiation [*Mlawer et al.,* 1997] and Dudhia's scheme for short wave radiation [*Dudhia,* 1989], and |
| | | 208 X 232 grid points (18 km grid spacing) | 0600 UTC December 8 to 1200 UTC December 9, 2011 | |
| | | 502 X 553 grid points (6 km grid spacing) | 0600 UTC December 8 to 1200 UTC December 9, 2011 | 6. Land-surface processes following the Noah land surface model (Noah LSM) [*Chen and Dudhia,* 2001; *Ek et al.,* 2003]. |
| | | 802 X 760 grid points (2 km grid spacing) | 0600 UTC December 8 to 1200 UTC December 9, 2011 | |

**Table 1.** List of observational data sets and horizontal grid dimensions, integration times, and WRF model physics applied in all
5    three dust storms cases (Pokharel, 2016; Pokharel et al., 2017b).



Yamada, 1974; Janjić, 2001), (iii) moist convection following the Betts-Miller-Janjić cumulus scheme (Betts, 1986; Betts and Miller, 1986; Janjic, 1994)- only for the simulations with 54 and 18 km resolution, (iv) cloud microphysical processes following the Thompson double-moment scheme (Thompson et al., 2004, 2006), (v) radiative processes following the Rapid Radiative Transfer Model for long wave radiation (Mlawer et al., 1997) and Dudhia's scheme for short wave radiation (Dudhia, 1989), and (vi) land-surface processes from the Noah land surface model (Noah LSM) (Chen and Dudhia, 2001; Ek et al., 2003) were applied in WRF simulations in each of three cases.

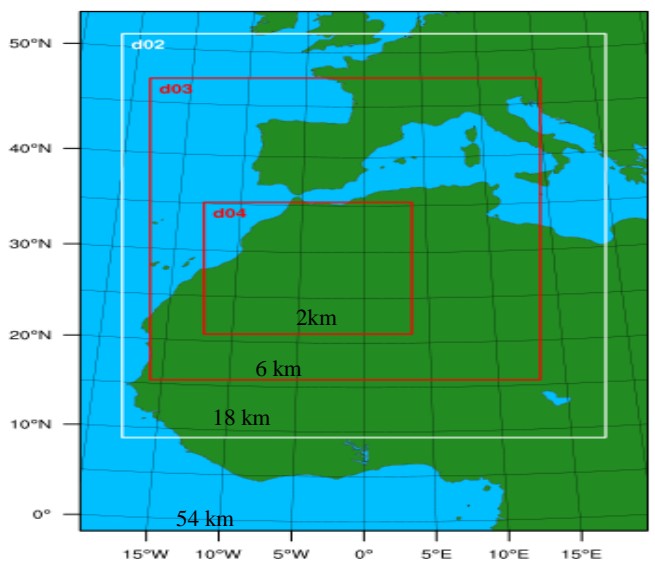

**Figure 3:** WRF domain configuration for the Harmatttan dust storm case in Figure 2 (Pokharel, 2016). do1, do2, do3, and do4 represent domains of 54, 18, 6, and 2 km resolution, respectively.

## 2. Results and discussion

### 2.1 Harmattan dust storm case study

#### 2.1.1 Observational and model analyses

Since Bechar lies to the east of Atlas Mountain and is the nearest sounding station to the Atlas Mountains, it is logical to expect that the information regarding the vertical profiles of meteorological variables in the lee of the mountains could be well represented in the analysis of before and after the Kelvin wave formation. The sounding of 0000 UTC March 2 shows that there was a thin inversion between 925-900 hPa (Figure 4a). Above this a dry adiabatic lapse rate between 900-585 hPa was present; again above this deep dry adiabatic layer, there was a presence of the shallow inversion layer, which was caused by a warm stable thermal ridge. These different kinds of vertical temperature lapse rates indicate the presence of the discontinuous stratification of the atmosphere over this area at that time and also shows the necessity of higher resolution data sets to reveal the detailed processes involved.

Surface archived observational data sets of Weather Underground at Tindouf (27°40′31″N 8°07′43″W) in Algeria showed that there was a major dust storm that occurred during the latter parts of March 2, 2004. This dust storm was associated with a surface north-northeasterly wind with a speed of 10 m/s after 2200 UTC on March 2, respectively. Besides this, Bechar (31°37′N 2°13′W, northeast of Tindouf), Adrar (27°52′N 0°17′W, east of Tindouf), and Timimoun (29°15′46″N 0°14′20″W, east of Tindouf) experienced reduced visibility consistent with increased atmospheric optical depth and dust from 1300 to 2200 UTC, 1400 to 2300 UTC, and 1300-1600 UTC on March 2, respectively (Pokharel et al., 2017b; Pokharel, 2016) (Figure 2).



5   NAAPS shows that at 1200 UTC of March 2 the evolution of the dust was over the northeast region of Algeria, including 20-28°N 15-8°E and over time the dust expanded broadly and occupied the 18-35°N 7°W-28°E region by 1200 UTC on March 3, 2004. MERRA-2 also shows the dust scattering AOD reached 0.25 over this region. This event signal in NAAPS is consistent with the surface observations described above.

MERRA data sets show that there was a positively-tilted trough (oriented northeast-southwest axis) in the southern part of Europe with an evolving ageostrophic wind indicative of the development of curved flow by the slight change of the wind direction (e.g. initiation of cyclonic curvature vorticity) near the trough axis at 28°N 1°W - 32°N 3°E at 0000 UTC on March 2, 2004 (Figures 5a and 5b). The geopotential height consistently showed a deep cold positively-tilted trough in time over the north to west part of Algeria. At 0600 UTC March 2 - 1200 UTC March 3, this deep trough cyclonically rotated to be oriented more northwest-southeast across Algeria (not shown) (Pokharel et al. 2017a and 2017b). The jet at 500 hPa was propagating towards the Atlas Mountains from north-northeast of Morocco and with further amplification over time, therefore it was continuously advancing southeastward influencing the trough till the time

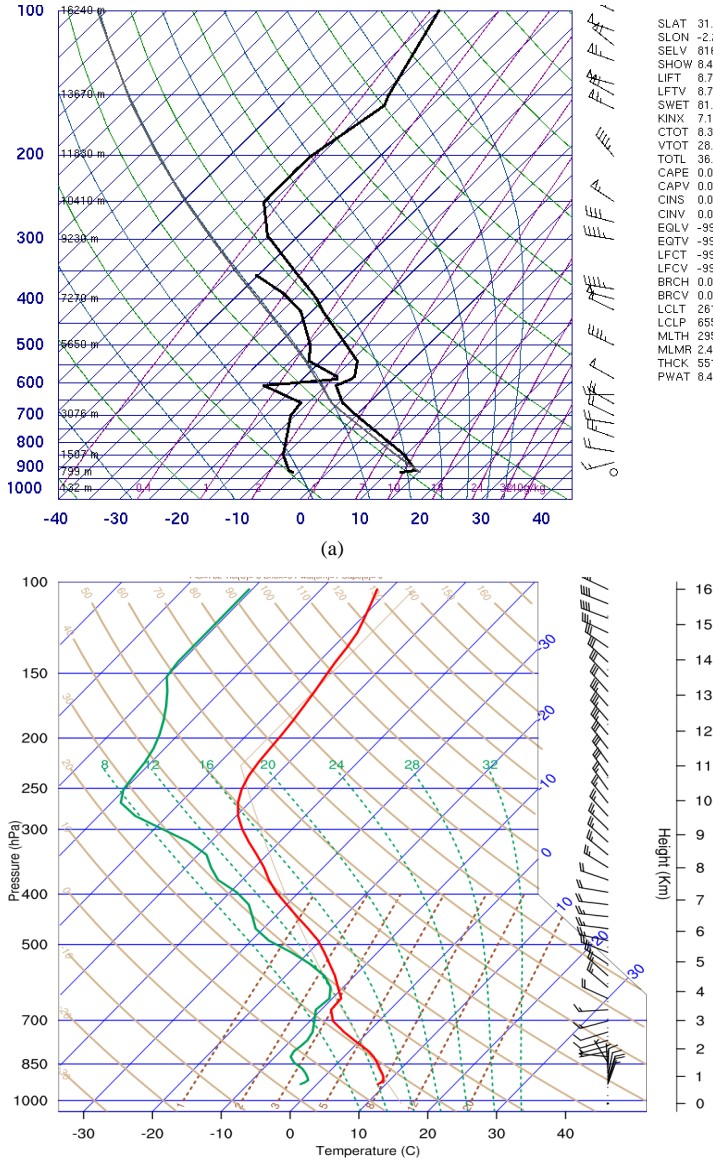

(a)

(b)

**Figure 4 a**. Atmospheric soundings at 0000 UTC March 2, 2004 at Bechar in Algeria (source: University of Wyoming). **b.** Atmospheric soundings at 0000 UTC March 2, 2004 at Bechar in Algeria plotted from WRF simulation (6 km resolution).





of the dust storm. The baroclinic amplification of the jet streak was consistent with the deepening of the trough at the meso-α/synoptic scale (between 200 km and 2000 km). Given the location of trough amplification of the exit region of the jet streak which was becoming proximate to the stable layers in Figures 4a and 4b, i.e., proximate to mountain-induced thermal perturbation on

the lee of the Atlas Mountains resulting in a mass perturbation (It is to be noted here that Figure 4a is observed sounding while 4b sounding is the product of WRF model. As both Figures are consistent with each other in regards to vertical profiles of temperature and wind WRF model outputs can be said to be validated). This interaction of the perturbed mass field ahead of the jet and the exit region of the jet leads to a breakdown of geostrophic balance. At 1800 UTC and

afterwards of March 2, the temperature and wind speed pattern at 925 hPa show the possible evolution and propagation of the Kelvin wave as it was propagating parallel to the Atlas Mountains in the southwestward direction ahead of the cold pool accompanying the large scale cold front (Figure 6a). All these processes will be analyzed with high resolution and time-continuous WRF numerical simulations.

### 2.1.2   WRF simulation analyses

Before entering in our specific analysis we would like to introduce a study of the jet adjustment processes, which are precursors of this large scale dust storm, discussed in recently published by Pokharel et al. (2017b). This jet adjustment processes were also one of the

precursors for the generation of Kelvin waves. Pokharel et al. (2016, 2017b) and Pokharel (2016) clearly show that there was an interaction of the exit region of the polar jet streak with a local thermally perturbed air mass on the leeward side of the Atlas Mountains, and summarizes the detail processes after this interaction as follows:  1) the generation of a jetlet at 0900 UTC on March 2, 2004 in the leeward side of the Atlas over the 30-32°N 6-2°W region ; 2) an occurrence

of the mass field adjustment processes modifying the wind field until it reaches to a new geostrophic balance; 3) in this adjustment process, a thermally direct transverse ageostrophic circulation in the exit region of the jetlet developed downstream from the mountain leading to the upward motion and the formation of the cold pool under the right exit of the jetlet (where velocity divergence exists; and 4) this cold pool led to the rise of the low-level high pressure

perturbation creating an ageostrophic/isallobaric wind as a return branch of the direct circulation of the exit region of the jetlet at the lower levels, i.e.,  925 hPa.


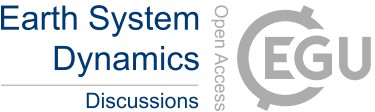
Following this finding, we analyzed our WRF simulated data sets further to see if there was any additional signal generated prior to or during the adjustment processes as discussed by Pokharel et al. (2017b) for the generation of the strong winds at the lower levels. We see that there was an additional meso-β scale mass field adjustment process shown by the geopotential height rise at 925 hPa at 1100 UTC and afterwards (Figures 7a and 7b). The temperature pattern at 0700 UTC and afterward shows the development of the cold region over 29-32°N 5°W-2°E reflecting the new mesoscale ascent zone.Within this cold pool, there was a comparatively colder region of air at lower levels than at upper levels in terms of the stretched vertical structure of the isentropic surfaces, indicative of the presence of the increased static stability of the atmosphere at this level (Figure 8a) (consistent with schematic Figure 1a).

This was the result of the presence of the blockage of the cold air column by the mountain range and generation of the initial mass impulse (Figures 8a and 8b) (consistent with schematic

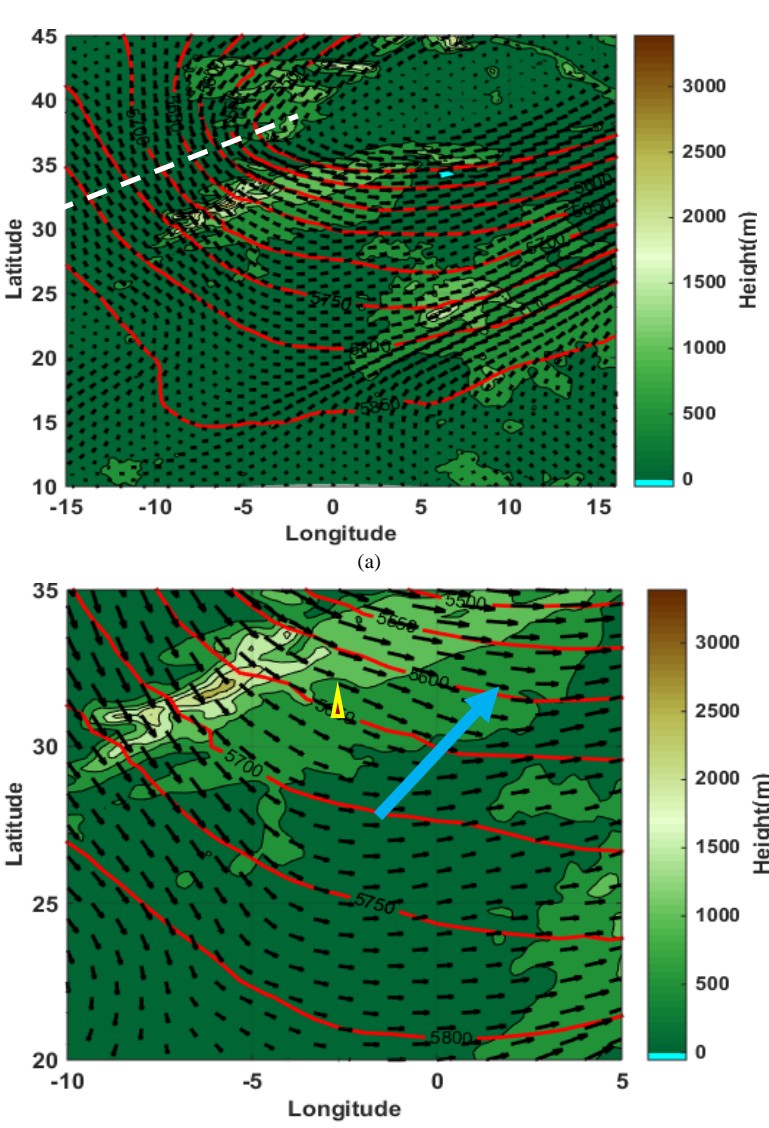

(a)

(b)

**Figure 5 a.** Geopotential height in steps of 50m (red contours), wind speed/direction(small black colored vectors), and the orientation of the trough at 0000 UTC on March 2, 2004 at the 500 hPa level from MERRA re-analysis with horizontal resolution of 54 km (Pokharel, 2016). The shading illustrates the orographic heights in meters. **b.** Enlargement of geopotential height in steps of 50 m (red contours), wind speed/direction (vectors), and axis of evolving ageostrophic wind (big arrow) at the 500 hPa level at 0000 UTC on March 2, 2004 from MERRA re-analysis with horizontal resolution of 54 km (Pokharel et al., 2017b). The shading illustrates the orographic heights in meter. A triangle of red (inside) and yellow (outside) shows the location of Bechar where the soundings were recorded.



Figures 1a and 1b). This mass buildup by the jet adjustment process in the lower levels resulted in wind flow parallel to the mountains at 925 hPa by the ageostrophic isallobaric wind as discussed by Pokharel et al. (2017b). This indicated the possible evolution of the Kelvin wave (Thomson, 1879; Wang, 2002; Tilley, 1990) on the south to northeast edge of the lee of the Atlas (29-32°N 5°W-2°E) at 1100 UTC on March 2 (Figures 8c and 8d ) (consistent with schematic Figures 1c, 1d, and 1e).

The generation of this Kelvin wave relies on the stable stratification for sustaining a gravitational/vertical restoring force in the form of a wave oscillation, followed by Coriolis force acting to induce a horizontal oscillation/restoring force in time in the presence of the blocking mountain barrier. Over time the Kelvin wave, which had a wavelength extending more than 100 km orthogonal to the Atlas Mountains, propagated southwestward of the Atlas as a northeasterly wind (Figures 8e and 8f) (consistent with schematic Figures 1d and 1e). During the passage of time this northeasterly wind further strengthened and increased the dust emission (consistent with schematic Figure 1f). Afterwards, when this cold northeasterly wind, which had already significant momentum, interacted with the warm air column of stretched isentropes, which were

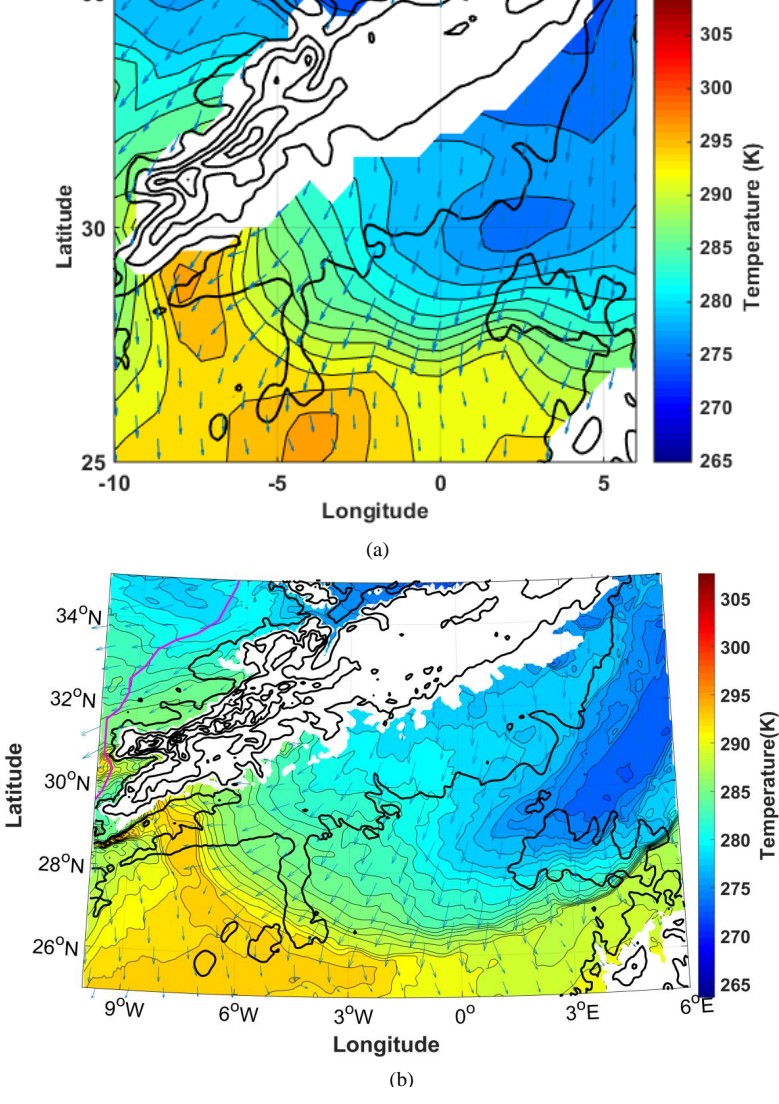

(a)

(b)

present south/southwestward of the leeside of the Atlas Mountains (Figure 8g) as discussed above, there was a large production of the dust over the region on the south/southwest side of the Atlas (Figure 8g) (consistent with schematic Figure 1f). This Kelvin wave started to organize the dust storms after the early weak dust storm caused by the downslope wind events as discussed by



Pokharel et al. (2017a). The Kelvin wave subsequently intensified the dust storm resulting in its growth upscale by first concentrating its energy in a narrow zone adjacent to the lee of the mountains in the afternoon on March 2 before it expanded over time as a suspension of dust in the atmosphere by the growing wind perturbation orthogonal to the Kelvin wave. It is also seen

that the northeasterly wind accompanying the Kelvin wave expanded further with the arrival of the Q-G cold pool from the region north/northeast of the Atlas Mountains parallelly (Figures 6b and 6c. Figure 6b is consistent with Figure 6a, which is a product of

observational data set as mentioned in the subsection 2.1.1). The presence of the low static stability on the south/southwestward

sections of the Atlas Mountains was shown by the stretching of the potential temperature in the vertical cross section in the

WRF simulations during dust storm genesis.

These processes illustrated that the development of the

northeasterly wind was associated with the huge cold surge and its interaction with the warm air column on the south and

southwestern edge of the Atlas Mountains as the Kelvin wave and quasi-geostrophic cold air surge merged in time. These

processes led to the generation of the large volume of dust from large areas on the leeward side (south and southwest of the

Atlas, 25-30°N 10°W-2°E) of the Atlas at around 2100 UTC on March 2 and afterwards. This is consistent with the

sequential dust storms at

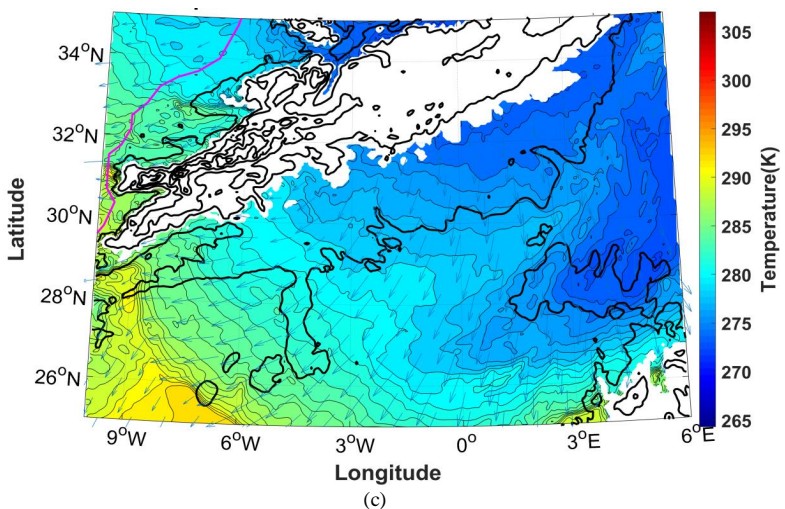

(c)

**Figure 6 a.** Temperature and wind speed/direction at 925 hPa on 1800 UTC March 2, 2004 (54 km resolution MERRA product) (Pokharel, 2016). **b.** Temperature and wind speed/direction at 925 hPa on 1800 UTC March 2, 2004 (6 km resolution WRF product). **c.** Temperature and wind speed/direction at 925 hPa on 2200 UTC March 2, 2004 (6 km resolution WRF product).

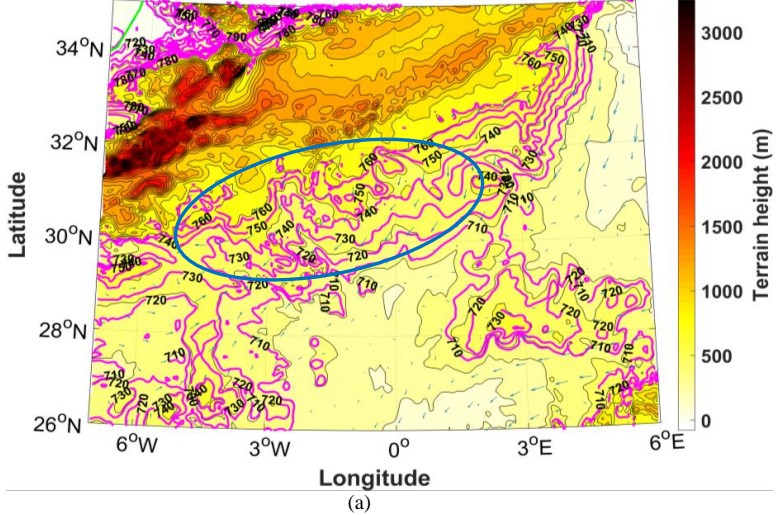

(a)

Tindouf, Bechar, and Adrar during this time period as discussed earlier in the sub-section 2.1.1.





Over time the WRF simulations and MERRA observations support the fact that the Kelvin wave propagated ahead of the cold pool accompanying the air behind the large scale cold front

(Figures 6a, 6b, and 6c). This led

the generation of the large volume of dust from the large areas on the leeward side (south and southwest of the Atlas, 25-31°N 10°W-2°E) of the Atlas at around 1800 UTC on March 2

and afterwards. This is also consistent with the dust storm at Tindouf at this time period. The presence of the area of warm air column on the south and

southwestern edge of the Atlas was also supported by the presence of the strong temperature gradient at the edge of this cold pool consistent with

the studies by Parmenter (1976); Garreaud and Wallace (1998); Liebmann et al. (1999). Here, an evolution of the Kelvin wave process as discussed earlier

is consistent with Tilley (1990), which states that Kelvin wave is the propagation of parallel wind

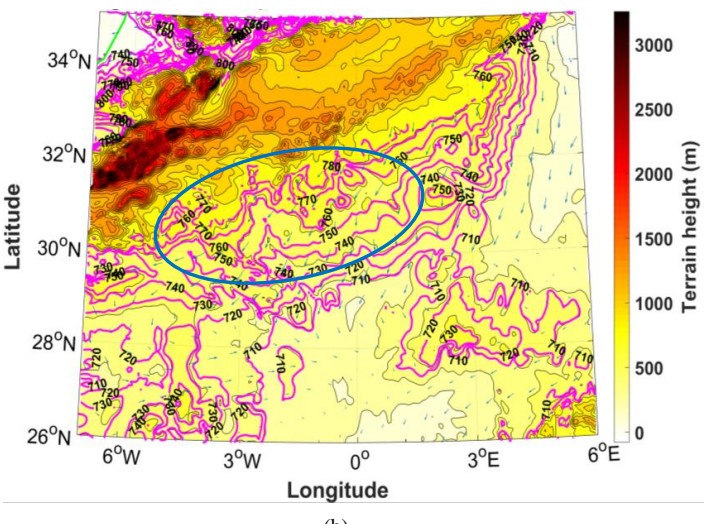

(b)

**Figure 7 a.** Geopotential height in steps of 5m (magneta contours) and wind speed/direction (small blue colored vectors) at 925 hPa on 0700 UTC March 2, 2004 (6 km resolution WRF product). Green circle indicates the region where the geopotential height increased over time. **b.** Geopotential height in steps of 5m (magneta contours) and wind speed/direction(small blue colored vectors) at 925 hPa on 1100 UTC March 2, 2004 (6 km resolution WRF product). Green circle indicates the region of geopotential height rises at 1500 UTC compared to 1200 UTC.

with the boundary on its right in the Northern hemisphere. The generation of Kelvin wave occurs when there is a shallow barotropic layer of air column (fluid) impinging on a physical barrier

with a vertical height > fluid depth. In such a case, the flow of this air column is blocked by the barrier creating a buildup mass in the area immediately adjacent to the barrier. After that, this excess mass is released as gravity waves and once the fluid has been set in motion the Coriolis force begins to act upon it, turning the air flow towards its right. Over time this process continues and the air flow as a Kelvin wave becomes directed parallel to and is right-bounded by the

barrier.













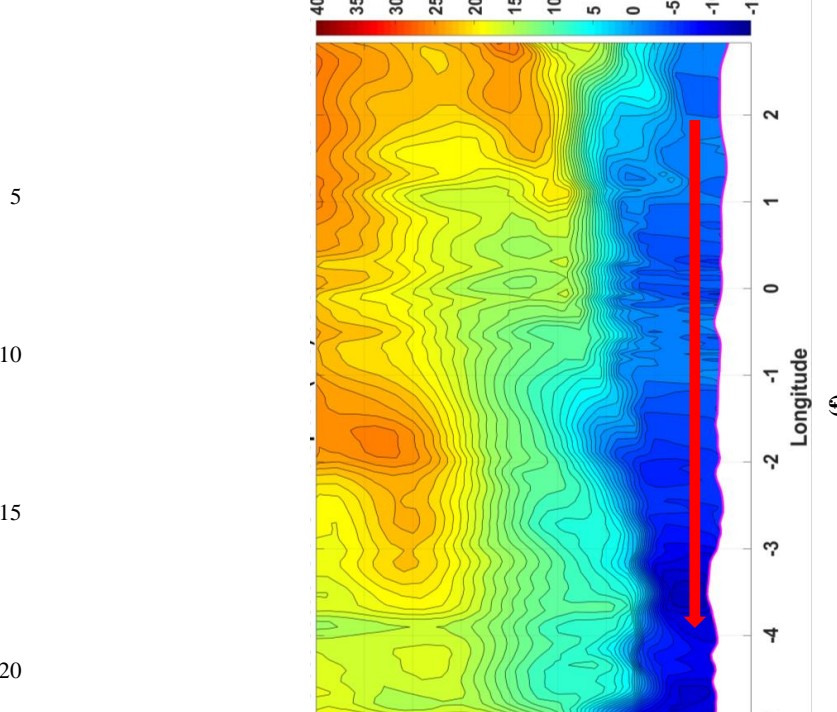

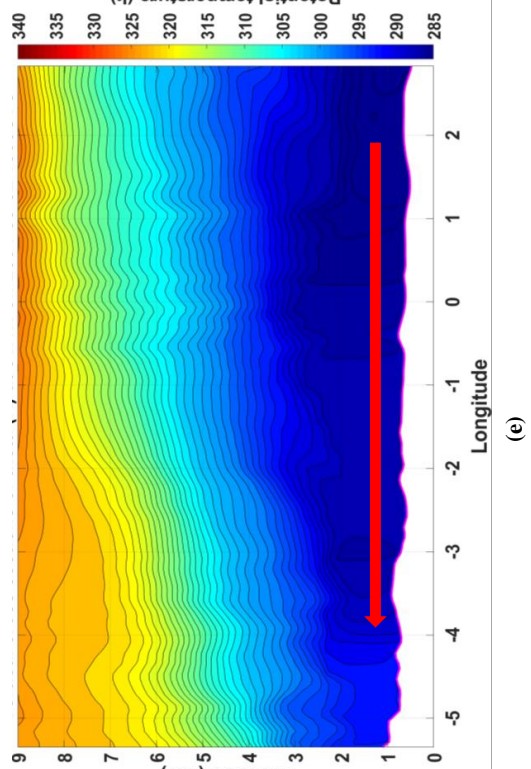





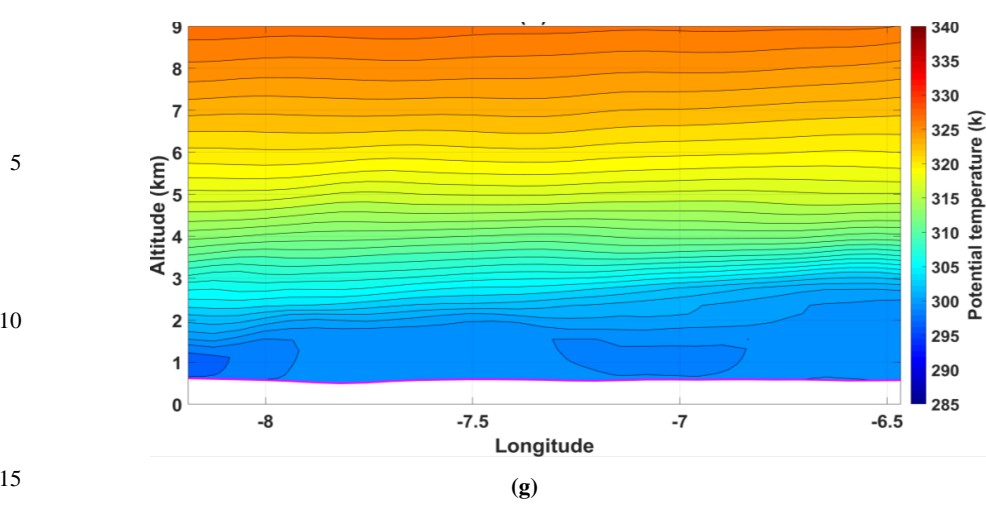

**(g)**

**Figure 8 a.** Vertical cross sections of potential temperature at 31.1˚ N on 0700 UTC March 2, 2004 (6 km resolution WRF product). Red circle indicates region of sinking of air column (blocking of air). **b.** Vertical cross sections of u-wind speed at 31.1˚N on 0700 UTC March 2, 2004 (6 km resolution WRF product). Red circle indicates region of blocking of air. **c.** Vertical cross sections of potential temperature at 31.1˚N on 1100 UTC March 2, 2004 (6 km resolution WRF product). Red arrow indicates generation of Kelvin waves (cold air surge). **d.** Vertical cross sections of u-wind speed component at 31.1˚ N on 1100 UTC March 2, 2004 (6 km resolution WRF product). Red arrow indicates generation of Kelvin waves. **e.** Vertical cross sections of potential temperature at 31.1˚N on 1500 UTC March 2, 2004 (6 km resolution WRF product). Red arrow indicates Kelvin wave (cold air surge) over time. **f.** Vertical cross sections of u-wind speed at 31.1˚N on 1500 UTC March 2, 2004 (6 km resolution WRF product). Red arrow indicates Kelvin wave over time. **g.** Vertical cross sections of potential temperature at 28.85˚N on 1500 UTC March 2, 2004 (6 km resolution WRF product).



### 2.2 Saudi dust storm case study

Surface stations of Saudi Arabia, such as Arar (30.90°N 41.13°E), Hafr al-Batin (28.33°N 46.12°E), Ha'il (27.43°N 41.68°E), Jeddah (21.70°N 39.18°E), Al-Madinah (24.55°N 39.71°E), Makkah (21.43°N 39.77°E), Rafha (29.63°N 43.48°E), Riyadh (24.65°N 46.64°E), and Turaif (31.69°N 38.73°E) observed dust storms in March 9, 2009 (Figure 9). We analyzed surface observational data, atmospheric soundings, MERRA data sets, and detailed WRF simulated high resolution data sets produced from WRF domain configuration (Figure 10) as discussed in the Harmattan case for determining the possible generation of Kelvin waves and its role to organize and lead the large scale dust transport in this particular case; but we could not see any significant signal of it as stated by Tilley (1990) in and around Saudi Arabia to cause

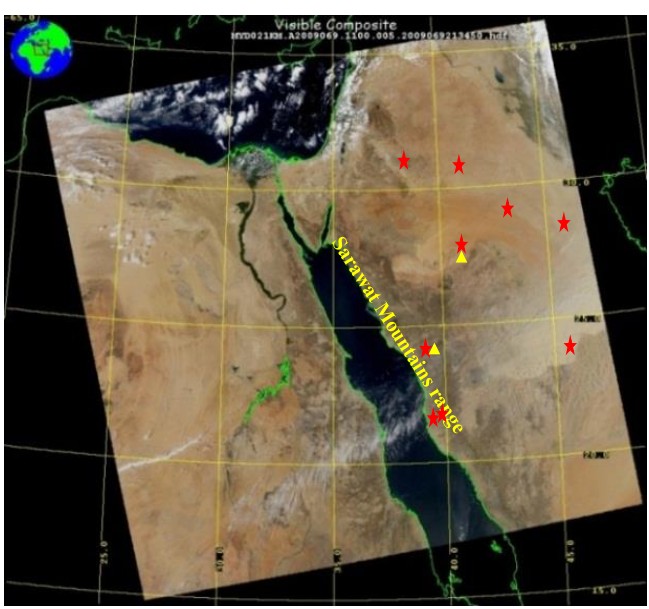

**Figure 9.** Saudi dust storm image captured by MODIS/Aqua at 1100 UTC on March 10, 2009 (source: https://ladsweb.nascom.nasa.gov). Red stars indicate surface stations (Adrar, Hafr al-Batin, Hail, Jeddah, Al-Madinah, Rafha, and Turaif in Saudi Arabia which captured the dust storms from the early of March 9, 2009 (source: wunderground.com). Yellow triangles indicate soundings stations at Hai'l and Al-Madinah in Saudi Arabia. (Pokharel et al., 2017b).

this widespread dust storm (Figure 9). Based on this condition, it is thought that it is not worthwhile to include all the detail analyses of the surface observations, soundings, and different model outputs here as discussed in the Harmattan case; so they are excluded in this manuscript.

### 2.3 Bodélé Depression dust storm case study

#### 2.3.1 Observational and model analyses

Though there were some observational stations in Chad, only one station, which is Ndjamena (12.13°N 15°E) (southeast of the Bodélé Depression) recorded dust storms with a northeasterly wind of 3 m/s speed at 0600 UTC on December 8. At 0600 UTC on December 8, 2011 (Figure 11), NAAPS shows the first evolution of the dust in the regions west/southwestward and equatorward of the Tibesti Mountains and after 6 hours (1200 UTC) the strength and the areal extension of the dust including the sulfate mineral increased and became concentrated mainly in the region southwest of the Tibesti Mountains as the dust-scattering AOD increased to 0.12 over this region as revealed by the MERRA-2.





MERRA data sets show the development of a positively tilted trough and the development of the ageostrophic wind consistent with the

initiation of the cyclonic vorticity near the trough axis at 19–20°N 17–19°E at 0600 UTC on 8 December (not shown). The positively tilted trough slightly deepened until the

occurrence of the dust emission. Although the strength of the positively tilted trough was weak compared to the other cases, falling heights still revealed the upper-air

disturbance over the northeast region of Niger, including north and northwestern regions of Chad which included Aozou, and southeast of Libya. From 0900 UTC 8 December

to 0900 UTC 9 December this trough propagated southeastward and this propagation of the trough along with the jet stream were both associated with quasi-geostrophic lifting ahead of the

trough and sinking behind it. A jet at 500 hPa was over the Tibesti Mountains and two additional jets were north of Chad from 0000 UTC 8 December. The jet, which was over the Tibesti, was coupled

to the trough until the occurrence of this dust storm. On the other hand, the Kelvin wave seemed to be evolving as it was flowing parallel to the Tibesti ahead of the large scale cold pool accompanying

the large scale cold front at 0800 UTC and afterwards shown by the temperature and wind speed pattern at the 850 hPa level at that time (Figure 12a). So, to analyze all these aforesaid conditions, we

again use high-resolution WRF simulations for the detailed dynamical development of a Kelvin Wave in this severe dust storm.


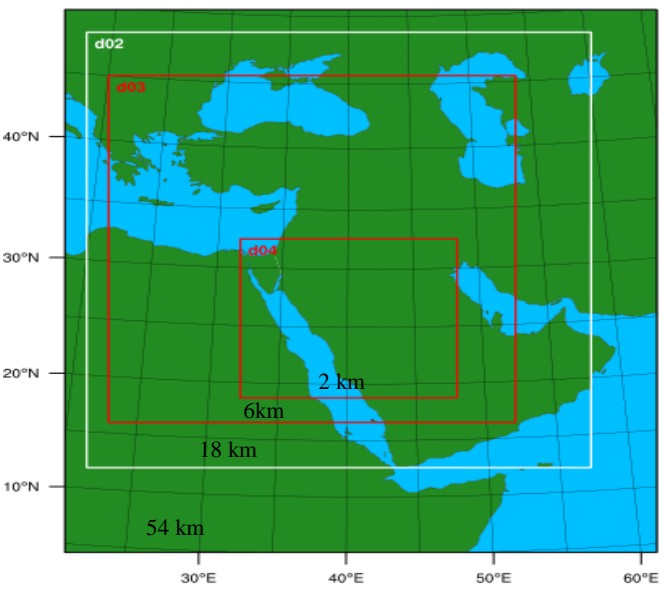

**Figure 10.** WRF domain configuration for the Saudi dust storm case shown in Figure 9 (Pokharel, 2016). do1, do2, do3, and do4 represent domains of 54, 18, 6, and 2 km resolution, respectively.

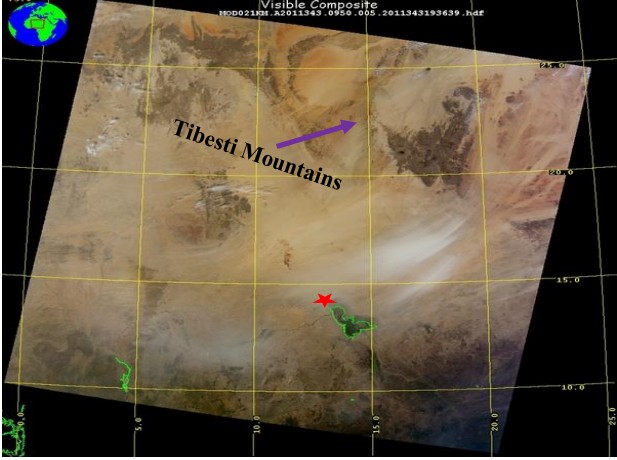

**Figure 11.** Bodélé Depression dust storm image captured by MODIS/Terra at 0950 UTC on December 9, 2011(source: https://ladsweb.nascom.nasa.gov). The red star indicates surface station at Ndjamena in Chad which captured the dust storms from 0600 UTC on December 8 to December 9, 2011 (source: wunderground.com) (Pokharel et al., 2017b). The x and y coordinates of this station are mentioned in observational and model subsection of 2..3.1 of Bodélé Depression dust storm case study.

**Figure 12 a.** Temperature and wind speed/direction at 850 hPa on 1800 UTC December 8, 2011(54 km resolution MERRA product). **b:** Temperature and wind speed/direction at 850 hPa on 1800 UTC December 8, 2011 (54 km resolution WRF product).



### 2.3.2   WRF simulation analyses

As mentioned above in the Harmattan case, Pokharel et al. (2017b) states in this case also that there was an interaction of the subtropical jet with the perturbed warm air mass on the leeward side (south/southwest/southeast) of the Tibesti Mountains that led to the different
processes (i.e. establishment of a meso-β scale adjustment process) as mentioned in the above Harmattan case.  In addition to this meso-β scale adjustment process in the lower levels, there was an additional interesting meso-β scale feature shown in the model output. At and after 0700 UTC on December 8 of 2011, as
discussed earlier in the Harmattan case we also see that there was an additional meso-β scale mass field adjustment process shown by the geopotential height rise at 925 hPa (Figures 14a and 14b).

Similarly, the temperature pattern at 925 hPa shows that there was a cold pool over the 19-30°N 10-30°E region. Within this wide cold region, there was a comparatively colder
region of air at the lower level than at upper levels in terms of the structure of the isentropic surfaces in the east/northeast side of the Tibesti Mountains (21-23°N 19-21°E) (Figure 15a) (consistent with schematic Figure 1a).
This cold column indicates the presence of the stability of the atmosphere at this level. This

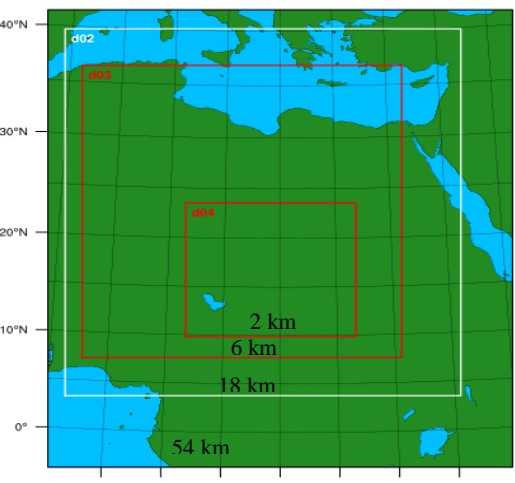

**Figure 13.** WRF domain configuration for the Bodélé Depression dust storms dust storm case shown in Figure 11 (Pokharel, 2016). do1, do2, do3, and do4 represent domains of 54, 18, 6, and 2 km resolution, respectively.

was the result of the presence of the blockage of the cold air column by the mountain range and generation of the initial mass impulse (Figures 15l and 15b) (consistent with schematic Figures 15a and 15b). From this buildup of mass by the jet adjustment process in the lower levels as
discussed earlier, there was wind flow parallel to the Tibesti Mountains and equatorward at 925 hPa (Figure 12b). After flowing parallel to the Tibesti Mountains this equatorward-directed wind rotated anti-cyclonically or southwestward along the east slope of the Tibesti Mountains as a northeasterly wind which strengthened in the time. When the wind flow at this low level stable region became parallel to these mountains, it can be inferred that there was a generation of the
Kelvin wave (Thomson, 1879; Tilley, 1990) (Figures 12a, 12b, 15c, and 15d) (consistent with schematic Figures 1c, 1d, and 1e). There was a close association between the subtropical jet streak imbalance (Pokharel et al., 2017b) and the Kelvin wave formation due to the presence of the inversion layer in the lee of the Tibesti Mountains. The presence of the cold pool at the bottom of the lee slope led to the pressure rise resulting in the wind flow parallel to the Tibesti
Mountains during the Kelvin wave formation at 925 hPa. This process was also supported by the generation of compensating jet generated by the low level pressure rise and its ageostrophic/isallobaric wind flow as discussed in the earlier case study. This Kelvin wave, which was trapped along lateral vertical mountain boundaries, was the result of the mass perturbations propagating parallel to the mountains barrier as discussed by Tilley (1990) and was
also representative of higher frequency form of adjustment process that might have played a vital role in generating this type of particular dust storm processes. It is important to note that when





**Figure 14 a.** Geopotential height in steps of 5m (magneta contours) and wind speed/direction (small blue colored vectors) at 925 hPa on 0700 UTC December, 8 2011 (6 km resolution WRF product). Green circle indicates the region where the geopotential height increased over time. **b.** Geopotential height in steps of 5m (magneta contours) and wind speed/direction (small blue colored vectors) at 925 hPa on 0900 UTC December, 8, 2011 (6 km resolution WRF product). Green circle indicates the region of geopotential height rises at 1500 UTC compared to 0700 UTC.









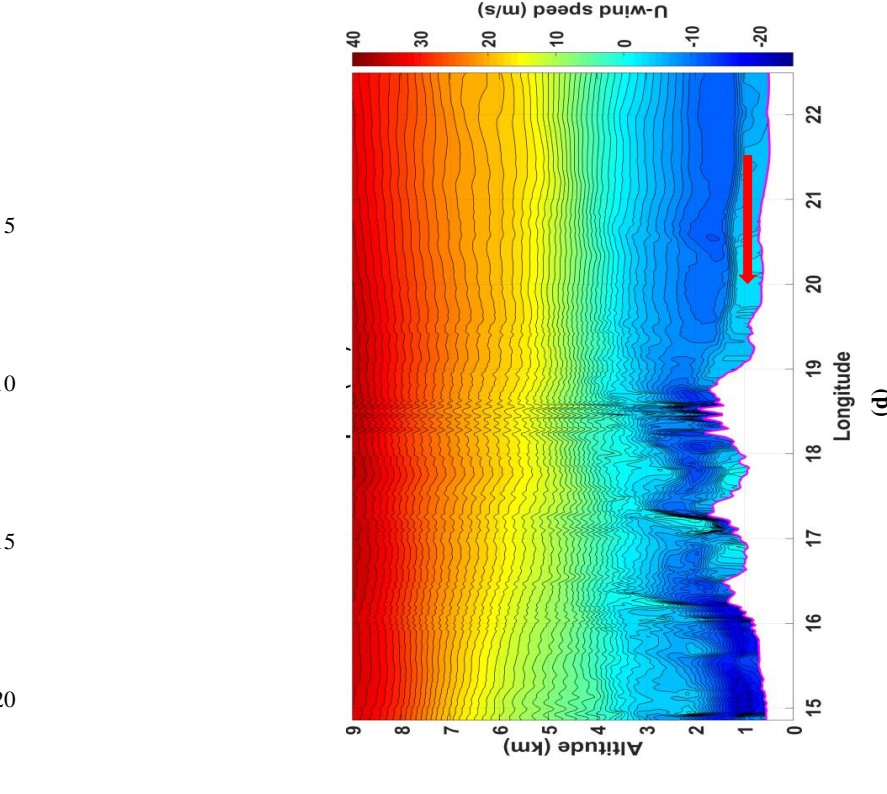

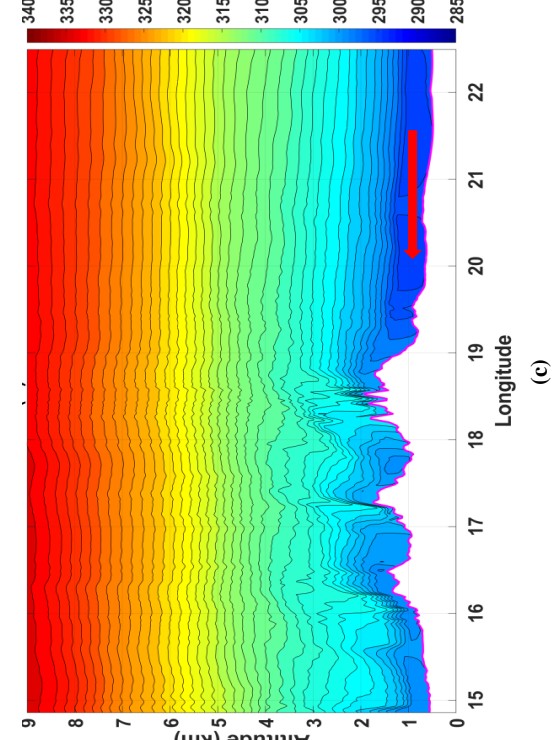

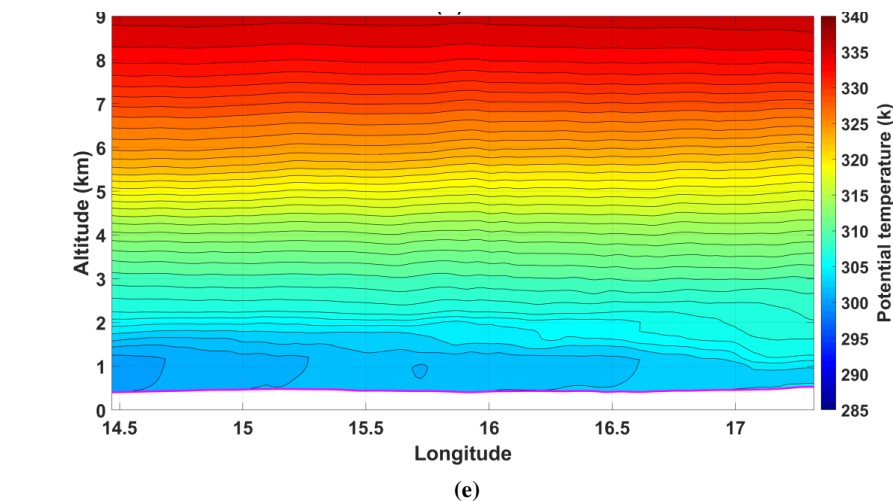

**(e)**

**Figure 15 a.** Vertical cross sections of potential temperature at 21.57° N on 0700 UTC December 8, 2011 (2 km resolution WRF product). Red circle indicates possible stability of the atmosphere. **b.** Vertical cross sections of u-wind speed at 21.57°N on 0700 UTC December 8, 2011 (2 km resolution WRF product). Red circle indicates blocking of the air. **c.** Vertical cross sections of potential temperature at 21.57°N on 1100 UTC December 8, 2011 (2 km resolution WRF product). Red arrow indicates Kelvin wave (cold air surge). **d.** Vertical cross sections of u-wind speed at 21.57° N on 1100 UTC December 8, 2011 (2 km resolution WRF product). Red arrow indicates Kelvin wave. **e.** Vertical cross sections of potential temperature at 18.89°N on 1500 UTC December 8, 2011 (2 km resolution WRF product).

this northeasterly wind became directed southwestward of the Tibesti Mountains, it accelerated. In time, this northeasterly momentum interacted with the warm air column resulting from the adiabatically-compressed downslope wind. This warm pool can be inferred from the strong gradient of temperature at 850 and 925 hPa and the vertically expansive isentropic surfaces at lower levels. This region was favorable for the generation of significant magnitudes of turbulence kinetic energy (TKE) from the contribution of wind shear and the buoyancy terms as defined in the TKE tendency equation suggested by (Stull, 2000),

$$\frac{\partial \text{TKE}}{\partial t} = -V.\nabla\text{TKE} + u*^2 \left(\frac{\partial u}{\partial z}\right) + g\left(\frac{Qs}{Tv}\right) - \varepsilon \qquad (1),$$

where $-V.\nabla TKE$ is TKE advection by the mean wind, $u_*^2(\partial u/\partial z)$ is generated shear, $g(Qs/Tv)$ is buoyancy, and $\varepsilon$ is the dissipation of TKE or eddy dissipation rate. This enhancement of the TKE developed a turbulent well-mixed circulation, which ablated the dust from the surface.

At 0900 UTC and onwards, the Kelvin wave mentioned in the above paragraph propagated ahead of the cold pool accompanying the northeasterly wind behind the large scale cold front from the north/northeast side of the Tibesti Mountains towards the south/southwest region of Chad, where the deep convective environment existed. The presence of the convective turbulent environment was consistent with the expansion of the isentropic surfaces (Figure 15e) and the strong gradient of the temperature at the edge of the cold pool at 850 and 925 hPa as suggested by Parameter (1976); Garneau and Wallace (1998); and Liebmann et al. (1999). The interaction of this cold pool accompanying the northeasterly wind and the highly warm air column created significant magnitudes of TKE (Figure 16) accompanying the generation of the turbulent eddies, which ablated the large volume of the diatomite dust from the surface, which is also consistent with Pokharel and Kaplan (2017). The surge of the cold pool over this region followed the early





enhancement of the warm air in the south/southwestward of the Tibesti is consistent with the description in Vizy and Cook (2009).

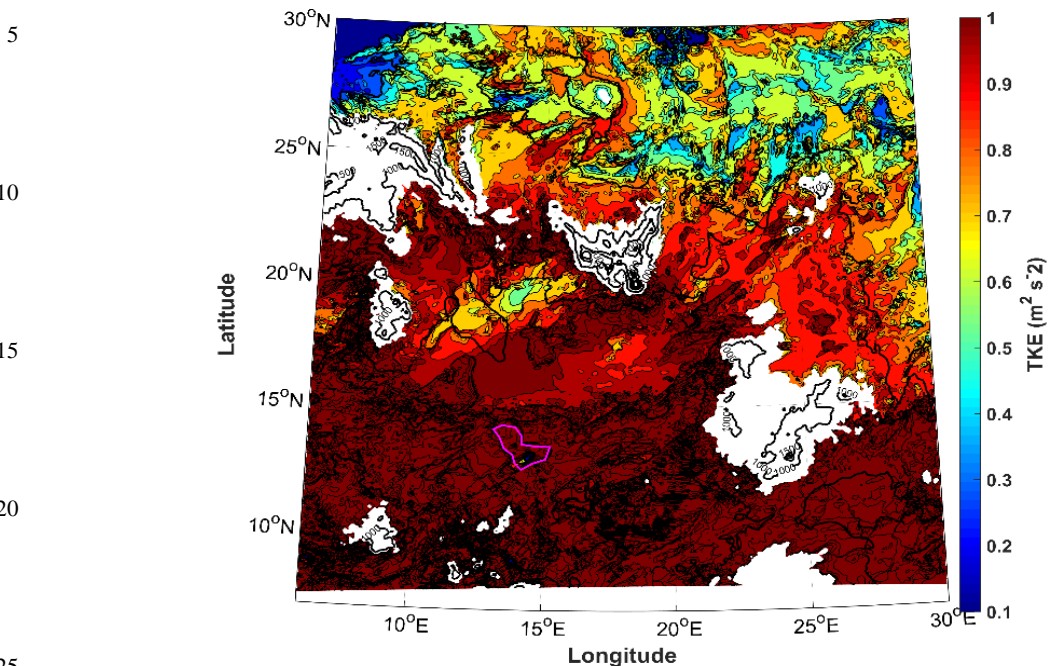

**Figure 16.** Turbulent kinetic energy at 925 hPa on 1200 UTC December 8, 2011 (6 km resolution WRF product).

### 3. Conclusions

Besides the downslope wind effects and the jet adjustment processes which were responsible to cause these three dust storms (Pokharel et al., 2017a, 2017b, and 2016; Pokharel, 2016) we found that the organization of the dust storms and the synoptic scale transport of dust from the Harmattan and Bodélé Depression dust storms were also seen to be caused by the Kelvin waves. The generation of Kelvin waves were facilitated by the jet adjustment processes in
the lee of the mountains and topographic blocking of the flow consistent with the lack of the sufficient kinetic energy of the air parcels to cross the mountains. This lack of kinetic energy to cross the mountains led to the 1) buildup of mass in the area immediately adjacent to respective mountains (consistent with schematic Figures 1a and 1b) , 2) release of excess mass from this build-up mass after certain period of time (consistent with schematic Figure 1c) , 3) effects of
Coriolis force on turning of the flow towards the right to achieve a balance (consistent with schematic Figure 1d), 4) wind flow accompanying Kelvin waves parallel to the mountains (consistent with schematic Figure 1e), 5) interaction of this wind accompanying Kelvin waves with the highly warm air column created significant magnitudes of TKE resulting in dust from the surface(consistent with schematic Figure 1f). This result is consistent with our generation of
Kelvin waves  hypotheses shown in schematic Figure 1 and also as discussed in the literature



reviews of this manuscript. This result contributes to a better understanding of how this type of large scale dust transport can be organized from this region to the U.S., Amazon basin, and Europe (which are shown by numerous previous studies) and might be due in part to Kelvin waves. We also think there is still a need to study major historical dust events that occurred in this region to confirm that these locations are suitable and are responsible for the generation of the Kelvin waves and the transport the dust on a regular basis.

**Author contributions**

Dr. Ashok Kumar Pokharel collected relevant data, performed the WRF model simulations and prepared this manuscript. Dr. Michael Kaplan edited and provided feedback on it.

**Competing interests**

The authors declare that they have no conflict of interest.

**Acknowledgments**

We would like to thank K.C. King, Robert David, Stephen Noble, Farnaz Hosseinpour, and Kacie Shourd for their help in collecting the observational data sets and running the WRF model. We would also like to thank to National Science Foundation (NSF) and National Center for Atmospheric Research (NCAR) for providing computational support.

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
