# Peer review of "Ashok Kumar Pokharel1, 2 and Michael L. Kaplan2, 3"

_Earth System Dynamics, 2019_

## Referee Comment (RC1) · Anonymous Referee #1 · 19 Jul 2019

An attempt is made to investigate the role of Kelvin waves in the development of dust storms for three cases involving orography, with the aid of WRF model simulations and satellite images. Despite the fact that I recognize that the investigation of this role is very interesting, my basic comment is that the analysis lacks of important evidence of the development of Kelvin waves. More specifically: 1. The authors support their statement on the temperature distribution of Figure 6 and geopontential in Figure 7 and vertical cross sections of potential temperature in Figure 8. First of all, these distributions are very messy and it is hard to recognize any patterns. Second, I think that the waves should be identified as streamline patterns or as geopotential anomalies or as anomalies of meridional wind. These plots are missing. Similar comment for the third case 2. Page 7, lines 30-35: The station locations should be displayed on the

map. Similarly with other locations referred in the manuscript 3. Section 2.2: I do not understand the reason for simply mentioning the finding that in the second case there is no evidence of development of Kelvin waves. The authors should get a better insight to investigate the reason for this, since the three cases are selected based on common criteria, involving the presence of orography. For instance, a first explanation could be related with the simulation of the case or with lack of data as compared to the other two cases. If not, the role of the orography is very likely to play a different role in this case. 4. I think that the structure of the paper should be modified. Section 3 should involve the Harmattan dust stom (3.1 Observational and model analysis, 3.2 WRF simulations) Section 4 should involve the second case and Section 3 the third case. 5. The conclusions are very poor and should be extended.

---

## Author Comment (AC1) · 3 Aug 2019

Dear Sir/Madam,

We hereby submit our revised manuscript "Organization of Dust Storms and Synoptic Scale Transport of Dust by Kelvin Waves" to the Earth System Dynamics journal with revisions to be considered for publication after incorporating our response regarding comments and suggestions provided by the 1st referee as follows:

Referee comments (RC1) An attempt is made to investigate the role of Kelvin waves in the development of dust storms for three cases involving orography, with the aid of WRF model simulations and satellite images. Despite the fact that I recognize that the investigation of this role is very interesting, my basic comment is that the analysis

[Figure]

lacks of important evidence of the development of Kelvin waves. More specifically: 1. The authors support their statement on the temperature distribution of Figure 6 and geopotential in Figure 7 and vertical cross sections of potential temperature in Figure 8. First of all, these distributions are very messy and it is hard to recognize any patterns. Second, I think that the waves should be identified as streamline patterns or as geopotential anomalies or as anomalies of meridional wind. These plots are missing. Similar comment for the third case 2. Page 7, lines 30-35: The station locations should be displayed on the map. Similarly with other locations referred in the manuscript 3. Section 2.2: I do not understand the reason for simply mentioning the finding that in the second case there is no evidence of development of Kelvin waves. The authors should get a better insight to investigate the reason for this, since the three cases are selected based on common criteria, involving the presence of orography. For instance, a first explanation could be related with the simulation of the case or with lack of data as compared to the other two cases. If not, the role of the orography is very likely to play a different role in this case. 4. I think that the structure of the paper should be modified. Section 3 should involve the Harmattan dust storm (3.1 Observational and model analysis, 3.2 WRF simulations) Section 4 should involve the second case and Section 3 the third case. 5. The conclusions are very poor and should be extended.

1 Referee comment: An attempt is made to investigate the role of Kelvin waves in the development of dust storms for three cases involving orography, with the aid of WRF model simulations and satellite images. Despite the fact that I recognize that the investigation of this role is very interesting, my basic comment is that the analysis lacks of important evidence of the development of Kelvin waves. More specifically: 1. The authors support their statement on the temperature distribution of Figure 6 and geopotential in Figure 7 and vertical cross sections of potential temperature in Figure 8. First of all, these distributions are very messy and it is hard to recognize any patterns. Second, I think that the waves should be identified as streamline patterns or as geopotential anomalies or as anomalies of meridional wind. These plots are missing. Similar comment for the third case.

Our response: We appreciate your comments and suggestion. We would like to clarify that Figures 6a-c presented here are showing the wind flow pattern, but not the temperature distribution. This wind flow, which is shown by the blue arrow, is parallel to the mountains - a fundamental character of Kelvin waves formed along the boundaries of topography. Please note that we have drawn the two major streamlines that define Kelvin wave motion in the Figures 6b and 6c of the revised manuscript. First, the acceleration parallel to the mountain with a slight right turn in response to the Coriolis force. Second the mountain-perpendicular acceleration in response to the rapid mass build-up consistent with leeside cooling. Note that in response to your issue with the complexity of the height fields in the previous Figure 7 we removed that figure as well as the similar Figure 14. The streamline additions are added on the caption of Figures 6b-c of the revised manuscript. Similarly, new Figures 7a and 7b of the revised manuscript show blockage of the cold air column by the mountain range and generation of the initial mass impulse (consistent with Figure 6b). New Figures 7c and 7d are showing evolution of Kelvin waves, a cold edge wave after the excess mass release and new Figures 7e and 7f depict Kelvin waves as a northeasterly wind signal directed towards the southwest of the Atlas Mountains. New Figure 7g depicts the vertical stretching of isentropic surfaces consistent with dynamical destabilization and hence deep mixing, i.e., a favorable region of dust emission after the generation of significant turbulence kinetic energy with the interaction of Kelvin waves (wind shear) (new Figures 7e and 7f) with buoyancy (new Figure 7g) thus resulting in TKE generation. For the 3rd case (Bodele Depression case) similar explanations are applied as discussed for the case 1 (new Figures 6 - 7). New Figures 10 and 11 of the revised manuscript are for the similar explanations for the third case.

2 Referee comment: Page 7, lines 30-35: The station locations should be displayed on the map. Similarly with other locations referred in the manuscript.

Our response: The station locations for two cases (Harmattan and Bodele Depression cases) are displayed on the maps of the revised manuscript (Figures 2b and 8b).

[Figure]

3 Referee comment: Section 2.2: I do not understand the reason for simply mentioning the finding that in the second case there is no evidence of development of Kelvin waves. The authors should get a better insight to investigate the reason for this, since the three cases are selected based on common criteria, involving the presence of orography. For instance, a first explanation could be related with the simulation of the case or with lack of data as compared to the other two cases. If not, the role of the orography is very likely to play a different role in this case.

Our response: We appreciate your concerns. We also think that it is not worthwhile to include the Saudi case in section 2.2 (in old manuscript) for the convenience of readers since we do not have unambiguous signals of Kelvin waves formation in this case though we carried out detail analyses with the help of observational and WRF model data sets as we did for the other two cases (Harmattan and Bodele Depression). So, we have removed section 2.2 in the revised manuscript. This information is also included in the conclusion of the revised manuscript. Please do understand that all validation of the Saudi Case indicated that the simulation was accurate but the dynamics not necessarily the same as the other two case studies.

4 Referee comment: I think that the structure of the paper should be modified. Section 3 should involve the Harmattan dust storm (3.1 Observational and model analysis, 3.2 WRF simulations) Section 4 should involve the second case and Section 3 the third case.

Our response: Yes, we have modified the structure in the revised manuscript.

5 Referee comment: The conclusions are very poor and should be extended.

Our response: Yes. We have extended the conclusions in the revised manuscript.

Sincerely, Ashok Kumar Pokharel
* * *
**Figure 2b:** Topographical map of Algeria (http://www.carte-algerie.com/carte-algerie/carte-topographique-algerie.jpg) in which the red star indicates surface stations (Bechar, Tindouf,Timimoun, and Adrar ) which experienced the dust storms on 2 March 2004 (source: wunderground.com). The blue colored triangle indicates soundings station at Bechar in Algeria.

**Fig. 1.**

Figure 6

[Figure]

(a)

**Fig. 2.**

[Figure]

**Figure 6 a.** Temperature and wind speed/direction at 925 hPa on 1800 UTC March 2, 2004 (54 km resolution MERRA product) (Pokharel, 2016). Blue arrow lines indicate the wind flow. **b.** Temperature and wind speed/direction at 925 hPa on 1800 UTC March 2, 2004 (6 km resolution WRF product). Blue arrow lines indicate wind flow. The red circled areas represent the turning of the one wind component to the right in response to the Coriolis force and another component was turning to the left away from the mountains farther north and east as it goes south-southwestwards. **c.** Temperature and wind speed/direction at 925 hPa on 2200 UTC March 2, 2004 (6 km resolution WRF product). Blue arrow lines indicate wind flow. The red circled areas represent the turning of the one wind component to the right in response to the Coriolis force and another component was turning to the left away from the mountains farther north and east as it goes south-southwestwards.

**Fig. 3.**

[Figure]

Figure 7

**Fig. 4.**

[Figure]

Fig. 5.

[Figure]

**Fig. 6.**

[Figure]

**(g)**

**Figure 7 a.** Vertical cross sections of potential temperature at 31.1˚ N on 0700 UTC March 2, 2004 (6 km resolution WRF product). Red circle indicates region of sinking of air column (blocking of air). **b.** Vertical cross sections of u-wind speed at 31.1˚N on 0700 UTC March 2, 2004 (6 km resolution WRF product). Red circle indicates region of blocking of air. **c.** Vertical cross sections of potential temperature at 31.1˚N on 1100 UTC March 2, 2004 (6 km resolution WRF product). Red arrow indicates generation of Kelvin waves (cold air surge). **d.** Vertical cross sections of u-wind speed component at 31.1˚N on 1100 UTC March 2, 2004 (6 km resolution WRF product). Red arrow indicates generation of Kelvin waves. **e.** Vertical cross sections of potential temperature at 31.1˚N on 1500 UTC March 2, 2004 (6 km resolution WRF product). Red arrow indicates Kelvin wave (cold air surge) over time. **f.** Vertical cross sections of u-wind speed at 31.1˚N on 1500 UTC March 2, 2004 (6 km resolution WRF product). Red arrow indicates Kelvin wave over time. **g.** Vertical cross sections of potential temperature at 28.85˚N on 1500 UTC March 2, 2004 (6 km resolution WRF product).

[Figure]

**Fig. 7.**

[Figure]

**Figure 8b:** Topographical map of Chad (**https://www.pinterest.com.au/pin/510877151474548010/**). The red star in this map shows surface station at Ndjamena in Chad which captured the dust storms from 0600 UTC on December 8 to December 9, 2011 (source: wunderground.com) (Pokharel et al., 2017b).

**Fig. 8.**

[Figure]

**Figure 10.** Temperature and wind speed/direction at 925 hPa at 1800 UTC on
December 8, 2011(6 km resolution WRF product). Blue arrow lines indicate the
wind flow. Blue circled areas indicate one wind component was flowing west-
southwest close to the mountains as it turns to the right and the other wind
component was turning to the left away from the mountains farther north and east
as it goes south-southwestwards.

**Fig. 9.**

[Figure]

Figure 11

Fig. 10.

[Figure]

**Fig. 11.**

[Figure]

**Figure 11 a.** Vertical cross sections of potential temperature at 21.57˚ N on 0700 UTC December 8, 2011 (2 km resolution WRF product). Red circle indicates possible stability of the atmosphere. **b.** Vertical cross sections of u-wind speed at 21.57˚N on 0700 UTC December 8, 2011 (2 km resolution WRF product). Red circle indicates blocking of the air. **c.** Vertical cross sections of potential temperature at 21.57˚N on 1100 UTC December 8, 2011 (2 km resolution WRF product). Red arrow indicates Kelvin wave (cold air surge). **d.** Vertical cross sections of u-wind speed at 21.57˚ N on 1100 UTC December 8, 2011 (2 km resolution WRF product). Red arrow indicates Kelvin wave. **e.** Vertical cross sections of potential temperature at 18.89˚N on 1500 UTC December 8, 2011 (2 km resolution WRF product).

**Fig. 12.**

---

## Referee Comment (RC2) · Anonymous Referee #2 · 23 Aug 2019

Review of the manuscript (ESD-2019-28): "Organization of dust storms and synoptic scale transport of dust by Kelvin waves" by A. K. Pokharel and M. L. Kaplan

General and specific comments:

This is an interesting work concerning the large-scale dust transport in the vicinity of mountain ranges and the organization of dust storms by Kelvin waves in sequence to evolving finer scale atmospheric processes. In my opinion, the authors have taken excellent attention into the evolution of different scales of motion and methodology details to infer the hypothesis proposed for examination. This study is unquestionably a commendable effort with appropriate schematics to demonstrate the atmospheric processes (blocking of air near the mountains and mesoscale geostrophic adjustments in conjunction with geostrophic imbalances) and dust transport by Kelvin waves di-

rected parallel to the mountain barriers. Relevant aspects of ESD are considered in this study with adequate figures and tables to substantiate the details. Also, the advanced modelling tools and datasets available are used for the analysis to substantiate the hypothesis proposed in this study.

In all sense, I am fully convinced with the arguments and rationale presented in the study, and I do not really see any pitfalls in the reporting. Therefore, I recommend the manuscript for publication with a few minor concerns in figures for better clarity to the readers, given in the following.

My minor concerns are:

1. Figures are little bit clumsy to decipher the details. Contours can be a bit smoother for clarity, and the colours could be lighter (e.g., Figure 6) 2. Section 2.1.1: Surface stations described in the study (lines 35-40) can be marked in Figure 3 similar to Figure 9. 3. For better reading in the vertical cross-section figures, locations can be marked with vertical line in tune with the text.

---

## Author Comment (AC2) · 27 Aug 2019

Dear Sir/Madam,

We hereby submit our following responses in regards to comments/suggestion of 2nd referee on "Organization of Dust Storms and Synoptic Scale Transport of Dust by Kelvin Waves" to be considered for publication in Earth System Dynamics Journal as follows:

Referee comments (RC2) General and specific comments: This is an interesting work concerning the large-scale dust transport in the vicinity of mountain ranges and the organization of dust storms by Kelvin waves in sequence to evolving finer scale atmospheric processes. In my opinion, the authors have taken excellent attention into the evolution of different scales of motion and methodology details to infer the hypothesis proposed for examination. This study is unquestionably a commendable effort with appropriate schematics to demonstrate the atmospheric processes (blocking of air near the mountains and mesoscale geostrophic adjustments in conjunction with geostrophic imbalances) and dust transport by Kelvin waves directed parallel to the mountain barriers. Relevant aspects of ESD are considered in this study with adeguate figures and tables to substantiate the details. Also, the advanced modelling tools and datasets available are used for the analysis to substantiate the hypothesis proposed in this study. In all sense, I am fully convinced with the arguments and rationale presented in the study, and I do not really see any pitfalls in the reporting. Therefore, I recommend the manuscript for publication with a few minor concerns in figures for better clarity to the readers, given in the following. My minor concerns are: 1. Figures are little bit clumsy to decipher the details. Contours can be a bit smoother for clarity, and the colours could be lighter (e.g., Figure 6) 2. Section 2.1.1: Surface stations described in the study (lines 35-40) can be marked in Figure 3 similar to Figure 9. 3. For better reading in the vertical cross-section figures, locations can be marked with vertical line in tune with the text.

1 Referee comment: My minor concerns are: 1. Figures are little bit clumsy to decipher the details. Contours can be a bit smoother for clarity, and the colours could be lighter (e.g., Figure 6).

Our response: We appreciate your comments/suggestion. Figures have been clarified by adding additional information on them (e.g. Figures 6a, 6b, and 6c of the revised manuscript). Actually, contours in Figures 6a, 6b, and 6c are temperature contours plotted to show the strong temperature gradient over the region of interest. For this lower contour values are taken, which produce line more granularity and less smoothing. This kind of contours are quite helpful to see the strong temperature difference and the strong wind flow. On the other hand if we plot smooth contours with the help of higher values of contours they produce contour lines with more smoothing that appears less jagged, but won't meet our expectation of seeing a strong temperature gradient.

**ESDD**
2 Referee comment: Section 2.1.1: Surface stations described in the study (lines 35-40) can be marked in Figure 3 similar to Figure 9.

Our response: We appreciate your suggestion. We have added new Figures 2b and 8b in the revised manuscript, and surface stations described in the study are marked on them.

3 Referee comment: For better reading in the vertical cross-section figures, locations can be marked with vertical line in tune with the text.

Our response: We appreciate your suggestion. We have marked locations in the vertical cross sections by the vertical lines to tune with the text (e.g. in new Figures 7a, 7b, 7g, 11a, 11b, and 11e of the revised manuscript).

Sincerely, Ashok Kumar Pokharel
Figure 201. Programmal imp of Augustation Imposure contrasteria concentration supervised and the Bechari, Thiotogrammal and Adrar U, which experienced the dust storms on 2 March 2004 (source: wunderground.com). The blue colored triangle indicates soundings station at Bechar in Algeria.

Fig. 1.
Figure 6

Fig. 2.

Interactive

comment

---

## Author Response (AR1)

Dear Sir/Madam,

We hereby submit our following revised manuscript (with track changes and without track changes manuscripts in separate pdf files) "Organization of Dust Storms and Synoptic Scale Transport of Dust by Kelvin Waves" to the Earth System Dynamics journal with revisions to be considered for publication after incorporating our response regarding comments and suggestions provided by the 1st and 2nd referees as follows:

**Referee comments (RC1)**

An attempt is made to investigate the role of Kelvin waves in the development of dust storms for three cases involving orography, with the aid of WRF model simulations and satellite images. Despite the fact that I recognize that the investigation of this role is very interesting, my basic comment is that the analysis lacks of important evidence of the development of Kelvin waves. More specifically: 1. The authors support their statement on the temperature distribution of Figure 6 and geopotential in Figure 7 and vertical cross sections of potential temperature in Figure 8. First of all, these distributions are very messy and it is hard to recognize any patterns. Second, I think that the waves should be identified as streamline patterns or as geopotential anomalies or as anomalies of meridional wind. These plots are missing. Similar comment for the third case 2. Page 7, lines 30-35: The station locations should be displayed on the map. Similarly with other locations referred in the manuscript 3. Section 2.2: I do not understand the reason for simply mentioning the finding that in the second case there is no evidence of development of Kelvin waves. The authors should get a better insight to investigate the reason for this, since the three cases are selected based on common criteria, involving the presence of orography. For instance, a first explanation could be related with the simulation of the case or with lack of data as compared to the other two cases. If not, the role of the orography is very likely to play a different role in this case. 4. I think that the structure of the paper should be modified. Section 3 should involve the Harmattan dust storm (3.1 Observational and model analysis, 3.2 WRF simulations) Section 4 should involve the second case and Section 3 the third case. 5. The conclusions are very poor and should be extended.

1    **Referee comment:** An attempt is made to investigate the role of Kelvin waves in the

development of dust storms for three cases involving orography, with the aid of WRF model simulations and satellite images. Despite the fact that I recognize that the investigation of this role is very interesting, my basic comment is that the analysis lacks of important evidence of the development of Kelvin waves. More specifically: 1. The authors support their statement on the temperature distribution of Figure 6 and geopotential in Figure 7 and vertical cross sections of potential temperature in Figure 8. First of all, these distributions are very messy and it is hard to recognize any patterns. Second, I think that the waves should be identified as streamline patterns or as geopotential anomalies or as anomalies of meridional wind. These plots are missing. Similar comment for the third case.

**Our response:** We appreciate your comments and suggestion. We would like to clarify that Figure 6 is presented here showing the wind flow pattern by the blue arrow and the temperature distribution by the contour lines, respectively. This wind flow, which is shown by the blue arrow, is parallel to the mountains - a fundamental character of Kelvin waves formed along the boundaries of topography. Please note that we have drawn the two major streamlines that define Kelvin wave motion in the Figures 6a, 6b, and 6c of the revised manuscript. First, the acceleration parallel to the mountain with a slight right turn in response to the Coriolis force. Second the mountain-perpendicular acceleration in response to the rapid mass build-up consistent with leeside cooling. This information is added in the revised manuscript. Note that in response to your issue with the complexity of the height fields in the previous Figures 7a and 7b we removed that figures as well as the similar Figures 14a and 14b. The streamline additions are added on the caption of Figures 6a, 6b, and 6c of the revised manuscript. Similarly, new Figures 7a and 7b of the revised manuscript show blockage of the cold air column by the mountain range and generation of the initial mass impulse. New Figures 7c and 7d are showing evolution of Kelvin waves, a cold edge wave after the excess mass release and new Figures 7e and 7f depict Kelvin waves as a northeasterly wind signal directed towards the southwest of the Atlas Mountains. New Figure 7g depicts the vertical stretching of isentropic surfaces consistent with dynamical destabilization and hence deep mixing, i.e., a favorable region of dust emission after the generation of significant turbulence kinetic energy with the interaction of Kelvin waves (wind shear) (new Figures 7e and 7f) with buoyancy (new Figure 7g) thus resulting in TKE generation.

For the 3[rd] case (Bodele Depression case) similar explanations are applied as discussed for the case 1 (new Figures 6 - 7). New Figures 10 and 11 of the revised manuscript are for the similar explanations for the third case.

**2    Referee comment:** Page 7, lines 30-35: The station locations should be displayed on the map. Similarly with other locations referred in the manuscript

**Our response:** The station locations for two cases (Harmattan and Bodele Depression cases) are displayed on the topographical maps of the revised manuscript (Figures 2b and 8b).

**3  Referee comment:** Section 2.2: I do not understand the reason for simply mentioning the finding that in the second case there is no evidence of development of Kelvin waves. The authors should get a better insight to investigate the reason for this, since the three cases are selected based on common criteria, involving the presence of orography. For instance, a first explanation could be related with the simulation of the case or with lack of data as compared to the other two cases. If not, the role of the orography is very likely to play a different role in this case.

**Our response:** We appreciate your concerns. We also think that it is not worthwhile to include the Saudi case in section 2.2 (in old manuscript) for the convenience of readers since we do not have unambiguous signals of Kelvin waves formation in this case though we carried out detail analyses with the help of observational and WRF model data sets as we did for the other two cases (Harmattan and Bodele Depression). So, we have removed section 2.2 in the revised manuscript. This information is also included in the conclusion of the revised manuscript. Please do understand that all validation of the Saudi Case indicated that the simulation was accurate but the dynamics not necessarily the same as the other two case studies.

**4  Referee comment:** I think that the structure of the paper should be modified. Section 3 should involve the Harmattan dust storm (3.1 Observational and model analysis, 3.2 WRF simulations) Section 4 should involve the second case and Section 3 the third case.

**Our response:** Yes, we have modified the structure in the revised manuscript.

**5    Referee comment:** The conclusions are very poor and should be extended.

**Our response:** Yes. We have extended the conclusions in the revised manuscript.

**Referee comments (RC2)**

General and specific comments: This is an interesting work concerning the large-scale dust transport in the vicinity of mountain ranges and the organization of dust storms by Kelvin waves in sequence to evolving finer scale atmospheric processes. In my opinion, the authors have taken excellent attention into the evolution of different scales of motion and methodology details to infer the hypothesis proposed for examination. This study is unquestionably a commendable effort with appropriate schematics to demonstrate the atmospheric processes (blocking of air near the mountains and mesoscale geostrophic adjustments in conjunction with geostrophic imbalances) and dust transport by Kelvin waves directed parallel to the mountain barriers. Relevant aspects of ESD are considered in this study with adequate figures and tables to substantiate the details. Also, the advanced modelling tools and datasets available are used for the analysis to substantiate the hypothesis proposed in this study. In all sense, I am fully convinced with the arguments and rationale presented in the study, and I do not really see any pitfalls in the reporting. Therefore, I recommend the manuscript for publication with a few minor concerns in figures for better clarity to the readers, given in the following. My minor concerns are: 1. Figures are little bit clumsy to decipher the details. Contours can be a bit smoother for clarity, and the colours could be lighter (e.g., Figure 6) 2. Section 2.1.1: Surface stations described in the study (lines 35-40) can be marked in Figure 3 similar to Figure 9. 3. For better reading in the vertical cross-section figures, locations can be marked with vertical line in tune with the text.

**1. Referee comment:** My minor concerns are: 1. Figures are little bit clumsy to decipher the details. Contours can be a bit smoother for clarity, and the colours could be lighter (e.g., Figure 6)

**Our response:** We appreciate your comments/suggestion. Figures have been clarified by adding additional information on them (e.g. Figures 6a, 6b, and 6c of the revised manuscript). Actually,

contours in Figures 6a, 6b, and 6c are temperature contours plotted to show the strong temperature gradient (i.e. strong temperature difference) over the region of interest. For this lower contour values are taken, which produce line more granularity and less smoothing. This kind of contours are quite helpful to see the strong temperature difference and the strong wind flow. On the other hand if we plot smooth contours with the help of higher values of contours they produce contour lines with more smoothing that appears less jagged, but won't meet our expectation of seeing a strong temperature gradient.

2 **Referee comment:** Section 2.1.1: Surface stations described in the study (lines 35-40) can be marked in Figure 3 similar to Figure 9.

**Our response:** We appreciate your suggestion. We have added new Figures 2b and 8b in the revised manuscript, and surface stations described in the study are marked on them.

3 **Referee comment:** For better reading in the vertical cross-section figures, locations can be marked with vertical line in tune with the text.

**Our response:** We appreciate your suggestion. To tune with the text, we have marked locations by the vertical lines ( i.e. up and down arrows) in the vertical cross section figures (e.g. in new Figures 7a, 7b, 7g, 11a, 11b, and 11e) of the revised manuscript.

Sincerely,

Ashok Kumar Pokharel

[revised manuscript text omitted]

(c)

**Figure 6 a.** Temperature and wind speed/direction at 925 hPa on 1800 UTC March 2, 2004 (54 km resolution MERRA product) (Pokharel, 2016). Blue arrow lines indicate the wind flow. The red circled areas represent the turning of the one wind component to the right in response to the Coriolis force and another component was turning to the left away from the mountains farther north and east as it goes south-southwestwards. **b.** Temperature and wind speed/direction at 925 hPa on 1800 UTC March 2, 2004 (6 km resolution WRF product). Blue arrow lines indicate the wind flow. The red circled areas represent the turning of the one wind component to the right in response to the Coriolis force and another component was turning to the left away from the mountains farther north and east as it goes south-southwestwards. **c.** Temperature and wind speed/direction at 925 hPa on 2200 UTC March 2, 2004 (6 km resolution WRF product). Blue arrow lines indicate the wind flow. The red circled areas represent the turning of the one wind component to the right in response to the Coriolis force and another component was turning to the left away from the mountains farther north and east as it goes south-southwestwards.

[Figure]

(a)

[revised manuscript text omitted]

**Figure 12 a.** Temperature and wind speed/direction at 850 hPa on 1800 UTC December 8, 2011 (54 km resolution MERRA product). b. Temperature and wind speed/direction at 850 hPa on 1800 UTC December 8, 2011 (54 km resolution WRF product).

**3.2.2   WRF simulation analyses**

As mentioned above in the Harmattan case, Pokharel et al. (2017b) states in this case also that there was an interaction of the subtropical jet with the perturbed warm air mass on the leeward side (south/southwest/southeast) of the Tibesti Mountains that led to the different processes (i.e. establishment of a meso-β scale adjustment process) as mentioned in the above Harmattan case.  In addition to this meso-β scale adjustment process in the lower levels, there was an additional interesting meso-β scale feature shown in the model output. At and after 0700 UTC on December 8 of 2011, as discussed earlier in the Harmattan case we also see that there was an additional meso-β scale mass field adjustment process shown by the geopotential height rise at 925 hPa ( not shown).

[Figure]

**Figure 9.** WRF domain configuration for the Bodélé Depression  dust storm case shown in Figure 11 (Pokharel, 2016). do1, do2, do3, and do4 represent domains of 54, 18, 6, and 2 km resolution, respectively.

Similarly, the temperature pattern at 925 hPa shows that there was a cold pool over the 19-30°N 10-30°E region. Within this wide cold region, there was a comparatively colder region of air at the lower level than at upper levels in terms of the structure of the isentropic surfaces in the east/northeast side of the Tibesti Mountains (21-23°N 19-21°E) (Figure 11a) (consistent with schematic Figure 1a). This cold column indicates the presence of the stability of the atmosphere at this level. This was the result of the presence of the blockage of the cold air column by the mountain range and generation of the initial mass impulse (Figures 11b and 11c) (consistent with schematic Figures 1a and 1b). From this buildup of mass by the jet adjustment process in the lower levels as discussed earlier, there was wind flow parallel to the Tibesti Mountains and equatorward at 925 hPa (Figure 10). After flowing parallel to the Tibesti Mountains this equatorward-directed wind rotated anti-cyclonically or southwestward along the east slope of the Tibesti Mountains as a northeasterly wind which strengthened in the time. When the wind flow at this low level stable region became parallel to these mountains, it can be inferred that there was a generation of the Kelvin wave (Thomson, 1879; Tilley, 1990) (Figures 10, 11c, and 11d) (consistent with schematic Figures 1c, 1d, and 1e). There was a close association between the subtropical jet streak imbalance (Pokharel et al., 2017b) and the Kelvin wave formation due to the presence of the inversion layer in the lee of the Tibesti Mountains. The presence of the cold pool at the bottom of the lee slope led to the pressure rise resulting in the wind flow parallel to the Tibesti Mountains during the Kelvin wave formation at 925 hPa. This process was also supported by the generation of compensating jet generated by the low level pressure rise and its ageostrophic/isallobaric wind flow as discussed in the earlier case study. This Kelvin wave, which was trapped along lateral vertical mountain boundaries, was the result of the mass perturbations propagating parallel to the mountains barrier as discussed by

Tilley (1990) and was also representative of higher frequency form of adjustment process that might have played a vital role in generating this type of particular dust storm processes. It is important to note that when

[Figure]

**Figure 10.** Temperature and wind speed/direction at 925 hPa at 1800 UTC on December 8, 2011(6 km resolution WRF product). Blue arrow lines indicate the wind flow. Blue circled areas indicate one wind component was flowing west-southwest close to the mountains as it turns to the right and the other wind component was turning to the left away from the mountains farther north and east as it goes south-southwestwards.

[Figure]

**Figure 14 a.** Geopotential height in steps of 5 m (magenta contours) and wind speed/direction (small blue coloured vectors) at 925 hPa on 0700 UTC December, 8 2011 (6 km resolution WRF product). Green circle indicates the region where the geopotential height increased over time. **b.** Geopotential height in steps of 5 m (magenta contours) and wind speed/direction (small blue coloured vectors) at 925 hPa on 0900 UTC December 8, 2011 (6 km resolution WRF product). Green circle indicates the region of geopotential height rise at 1500 UTC compared to 0700 UTC.

[Figure]

[Figure]

[Figure]

[Figure]

[Figure]

(e)

**Figure 11~5~ a.** Vertical cross sections of potential temperature at 21.57° N on 0700 UTC December 8, 2011 (2 km resolution WRF product). Red arrows (up and down) indicate the depth of region of possible stability of the atmosphere. . **b.** Vertical cross sections of u-wind speed at 21.57°N on 0700 UTC December 8, 2011 (2 km resolution WRF product). Red arrows (up and down) indicate blocking of the air.  **c.** Vertical cross sections of potential temperature at 21.57°N on 1100 UTC December 8, 2011 (2 km resolution WRF product) Red arrow indicates Kelvin wave (cold air surge).  **d.** Vertical cross sections of u-wind speed at 21.57° N on 1100 UTC December 8, 2011 (2 km resolution WRF product). Red arrow indicates Kelvin wave.  **e.** Vertical cross sections of potential temperature at 18.89°N on 1500 UTC December 8, 2011 (2 km resolution WRF product). Red arrows (up and down) indicate warm air column of vertical stretch of isentropes.

this northeasterly wind became directed southwestward of the Tibesti Mountains, it accelerated. In time, this northeasterly momentum interacted with the warm air column resulting from the adiabatically-compressed downslope wind. This warm pool can be inferred from the strong gradient of temperature at 850 and 925 hPa and the vertically expansive isentropic surfaces at lower levels. This region was favorable for the generation of significant magnitudes of turbulence kinetic energy (TKE) from the contribution of wind shear and the buoyancy terms as defined in the TKE tendency equation suggested by (Stull, 2000),

$$\frac{\partial \text{TKE}}{\partial \text{t}} = -V.\nabla\text{TKE} + u*^2 \left(\frac{\partial u}{\partial z}\right) + g \left(\frac{Qs}{Tv}\right) - \varepsilon \qquad (1),$$

where $-V.\nabla TKE$ is TKE advection by the mean wind, $u_*^2 (\partial u/\partial z)$ is generated shear, $g (Qs/Tv)$ is buoyancy, and $\varepsilon$ is the dissipation of TKE or eddy dissipation rate. This enhancement of the TKE developed a turbulent well-mixed circulation, which ablated the dust from the surface.

At 0900 UTC and onwards, the Kelvin wave mentioned in the above paragraph propagated ahead of the cold pool accompanying the northeasterly wind behind the large scale cold front from the north/northeast side of the Tibesti Mountains towards the south/southwest region of Chad, where the deep convective environment existed. The presence of the convective turbulent environment was consistent with the expansion of the isentropic surfaces (Figure 11e) and the strong gradient of the temperature at the edge of the cold pool at 850 and 925 hPa as suggested by Parameter (1976); Garneau and Wallace (1998); and Liebmann et al. (1999). The interaction of this cold pool accompanying the northeasterly wind and the highly warm air column created significant magnitudes of TKE (Figure 12) accompanying the generation of the turbulent eddies, which ablated the large volume of the diatomite dust from the surface, which is also consistent with Pokharel and Kaplan (2017). The surge of the cold pool over this region followed the early enhancement of the warm air in the south/southwestward of the Tibesti is consistent with the description in Vizy and Cook (2009).

[Figure]

**Figure 12.** Turbulent kinetic energy at 925 hPa on 1200 UTC December 8, 2011 (6 km resolution WRF product).

**4. Conclusions**

Besides the downslope wind effects and the jet adjustment processes which were responsible to cause these three dust storms (Pokharel et al., 2017a, 2017b, and 2016; Pokharel, 2016) we found that the organization of the dust storms and the synoptic scale transport of dust from the Harmattan and Bodélé Depression dust storms were also seen to be caused by the Kelvin waves. Though we also carried out a detail analyses of Saudi dust storm case with the help of observational and WRF model data sets as we did for Harmattan and Bodélé Depression dust storm cases, for the convenience of readers we have not included the detail analyses of Saudi case in this manuscript since we do not have unambiguous signals of Kelvin waves formation in this case. It is also to be understood that all validation of the Saudi case indicated that the simulation was accurate but the dynamics not necessarily the same as the other two case studies.

[revised manuscript text omitted]